# Comparative transcriptomics of *Diuraphis noxia* and *Schizaphis graminum* fed wheat plants containing different aphid-resistance genes

Lina Aguirre Rojas[1], Erin Scully[2], Laramy Enders[3], Alicia Timm[4], Deepak Sinha[1,5], Charles Michael Smith[1] *

1 Department of Entomology, Kansas State University, Manhattan, KS, United States of America, 2 Stored Product Insect and Engineering Unit, USDA-ARS Centerfor Grain and Animal Health Research, Manhattan, KS, United States of America, 3 Department of Entomology, Purdue University, West Lafayette, IN, United States of America, 4 Department of Bioagricultural Sciences and Pest Management, Colorado State University, Fort Collins, CO, United States of America, 5 SAGE University, Indore, India

* cmsmith@ksu.edu

**Data Availability Statement:** All transcripts containing protein coding genes have been submitted to NCBI's Transcriptome Shotgun Assembly database. The entire transcriptome

## Abstract

The molecular bases of aphid virulence to aphid crop plant resistance genes are poorly understood. The Russian wheat aphid, *Diuraphis noxia*, (Kurdjumov), and the greenbug, *Schizaphis graminum* (Rondani), are global pest of cereal crops. Each species damages barley, oat, rye and wheat, but *S. graminum* includes fescue, maize, rice and sorghum in its host range. This study was conducted to compare and contrast the transcriptomes of *S. graminum* biotype I and *D. noxia* biotype 1 when each ingested phloem from leaves of varieties of bread wheat, *Triticum aestivum* L., containing no aphid resistance (*Dn0*), resistance to *D. noxia* biotype 1 (*Dn4*), or resistance to both *D. noxia* biotype 1 and *S. graminum* biotype I (*Dn7*, wheat genotype 94M370). Gene ontology enrichments, k-means analysis and KEGG pathway analysis indicated that 94M370 plants containing the *Dn7 D. noxia* resistance gene from rye had stronger effects on the global transcriptional profiles of *S. graminum* and *D. noxia* relative to those fed *Dn4* plants. *S. graminum* responds to ingestion of phloem sap from 94M370 plants by expression of unigenes coding for proteins involved in DNA and RNA repair, and delayed tissue and structural development. In contrast, *D. noxia* displays a completely different transcriptome after ingesting phloem sap from *Dn4* or 94M370 plants, consisting of unigenes involved primarily in detoxification, nutrient acquisition and structural development. These variations in transcriptional responses of *D. noxia* and *S. graminum* suggest that the underlying evolutionary mechanism(s) of virulence in these aphids are likely species specific, even in cases of cross resistance.

assembly containing both coding and non-coding genes has been deposited in USDA's Ag Data Commons at https://data.nal.usda.gov/dataset/de-novo-transcriptome-assembly-schizaphis-gramium-biotype-i-feeding-wheat and an annotated assembly of the transcripts that code for proteins is available at NCBI's Transcriptome Shotgun Assembly (TSA) database under the accession GIML00000000. Partial mitochondrial sequences for COI that were used to confirm the taxonomic identity of each aphid species are deposited in GenBank under MT011383 for S. graminum I and MN994435 for *D. noxia* biotype 1.

**Funding:** This work was supported by the Kansas Wheat Alliance and the Kansas State Agricultural Experiment Station and awarded to CMS. The funders had no role in study design, data collection and analysis, decision to publish, or preparation of the manuscript.

**Competing interests:** The authors have declared that no competing interests exist.

## Introductory note

The original version of this article was retracted [1] by the corresponding author and PLoS ONE Editors on January 10, 2018, after determination that samples referenced as *Diuraphis noxia* U.S. biotype 2 were instead *Schizaphis graminum* biotype I. This error came to light after unexpected results were obtained in subsequent PCR experiments using residual samples from [2]. The identities of the two aphid species actually used in the study were verified as described in the Materials and Methods (see below). The following manuscript replaces the retracted publication, using corrected aphid species information, the *D. noxia* biotype 2 genome assembly version WGS Accession JOTR00000000.1 [3], and new data regarding the reaction of *S. graminum* biotype I to wheat plants containing the *Dn4* or *Dn7 D. noxia* resistance genes.

## Introduction

Arthropods exhibit remarkable genetic plasticity in adapting to stresses posed by both abiotic and biotic factors. Insect crop pests have demonstrated the ability to express resistance to virtually all insecticides and virulence against the majority of plant genes controlling insect resistance [4,5]. Many species of aphid pests are virulent to aphid resistance genes in crop plants, providing them with protection from plant defenses [6]. Virulent strains of aphids, often referred to as biotypes, are defined as populations within a species that differ in their ability to feed successfully on particular plant genotypes [7]. Aphid biotypes are routinely detected by assessing the phenotypic reactions of plant varieties possessing different arthropod resistance genes to an arthropod population [8]. The interaction of resistance genes in the plant determine the virulence or avirulence of an aphid biotype to a plant resistance gene. However, beyond these phenotypic measures, the molecular bases of aphid virulence continue to be poorly understood.

Knowledge generated to date indicates that effector proteins present in the saliva of avirulent aphids are recognized by the defense response systems of insect-resistant plants, initiating the production of plant allelochemical defenses such as alkaloids, ketones, and organic acids [5] that prohibit an aphid from damaging or infesting the plant. Virulent aphids are thought to overcome normally resistant plant genes by release of suppressor proteins to mask aphid effectors from plant perception [9,10]. Several mechanisms have been proposed to explain how aphid virulence is mediated [11]. Enzymatic components in the salivary glands or midgut of some aphid species interfere directly with plant allelochemical defenses via detoxification or inhibition [12]. Some biotypes of the pea aphid, *Acyrthosiphon pisum* (Harris), exhibit variation in gene sequence and expression level that may influence host plant recognition and specialization [13].

The Russian wheat aphid, *Diuraphis noxia*, (Kurdjumov) has invaded all continents producing bread wheat, *Triticum aestivum* L., [14,15], and is expected to spread further into Asia, Europe, North and South America, and New Zealand [16]. Similarly, the greenbug, *Schizaphis graminum* Rondani, is a major global pest of bread wheat and sorghum, *Sorghum bicolor* L. In the United States, the greatest *S. graminum*—related losses occur in the Southern Great Plains, causing annual yield losses estimated at ~$250 million [17].

Fourteen *Gb* (greenbug) genes for resistance to *S. graminum*, and 14 *Dn* (*D. noxia*) genes for resistance to *D. noxia* have been identified from wild relatives of bread wheat, *Triticum aestivum*, or rye, *Secale cereale* L. [18,19]. Significant yield losses from both pests persist, despite the deployment of several of these genes in varieties of wheat resistant to each aphid [20,21]. Several *S. graminum* biotypes exist in wheat, sorghum and lawn and pasture grasses [22] and currently there are nine characterized biotypes of *D. noxia* in the U. S. and South Africa ([23,24].

The perpetual occurrence of aphid virulence to plant resistance genes necessitates an improved understanding of the molecular bases of virulence in order to better defend 21$^{st}$ century food crops from aphid-induced yield losses. Therefore, it was pertinent to investigate the impacts of different wheat varieties carrying either no resistance genes (*Dn0*), resistance to *D. noxia* biotype 1 (*Dn4*), or resistance to *D. noxia* biotype 1 and *S. graminum* (*Dn7*) on life history and transcriptomes of *D. noxia* and *S. graminum*. Our objectives were to confirm whether lines carrying the *Dn4* and *Dn7* genes also had cross-resistance to *S. graminum* and determine whether lines carrying these resistance genes had similar impacts on the transcriptomes of *D. noxia* and *S. graminum*.

## Materials and methods

### Insect and plant material

*Diuraphis noxia* biotype 1 aphids were collected from wheat fields near Hays, KS (38.8794˚ N, 99.3222˚ W). *Schizaphis graminum* biotype I originated from a field population on the Kansas State University campus in Manhattan, KS (39.188307˚ N, -96.605864˚ W). Neither field collection involved endangered or protected species. No specific permissions were required for these collections, as they were activities agreed upon by USDA-ARS scientists and scientists at Colorado State University and Kansas State University as a part of the Areawide Pest Management for Wheat: Management of Greenbug and Russian Wheat Aphid. The identity of each aphid was confirmed by PCR amplification of DNA from whole bodies and sequencing of a region of mitochondrial cytochrome c oxidase I (COI) from the PCR product. Partial COI sequences of the *S. graminum* I colonies used in these studies have been deposited at GenBank under MT011383. Partial COI sequences for *D. noxia* biotype 1 used in these studies have been deposited at GenBank under accessions MN994435. COI has been used effectively to identify both *Diuraphis noxia* and *Schizaphis graminum* [25]. Fresh DNA samples of three additional aphid species (*Sitobion avenae*, *Rhopalosiphum padi*, *Melanaphis sacchari*), were also amplified, along with archived and fresh DNA of *D. noxia* from Hungary, Spain, and North America (biotypes 1, 2, 4, 6, and 8). After COI identification, biotype identification and validation were independently performed for both *D. noxia* and *S. graminum* by plant differential diagnoses [26,27,28] at Stillwater, OK, and Manhattan, KS. Each aphid species was maintained in separate growth chambers at Kansas State University on the susceptible wheat cultivar 'Jagger.' Specimen samples (*S. graminum* biotype I voucher specimen #155, *D. noxia* biotype 1 voucher specimen #176) are deposited at the Museum of Entomological and Prairie Arthropod Research at Kansas State University.

The wheat varieties Yuma, containing no resistance genes (*Dn0*); the *D. noxia* biotype 1-resistant variety Yumar, containing the *Dn4* resistance gene [29]; and the variety 94M370, containing the *Dn7* gene for resistance to *D. noxia* biotype 2 [30] were used in experiments to compare the transcriptomes of *S. graminum* biotype I and *D. noxia* biotype 1. Yuma was developed from crosses between the *D. noxia*-susceptible wheat varieties NS14, NS25 and Vona. Yumar wheat was selected from a cross between Yuma and wheat plant introduction (PI) 372129, the source of *Dn4* [31]. The *Dn7* gene originates from the terminal region of the short arm of chromosome 1 (1RS) of rye, *Secale cereale* L., variety Turkey 77. *Dn7* resistance was transferred to Gamtoos wheat by a translocation of the rye 1RS segment into the short arm of wheat chromosome 1B [30,32], resulting in the breeding line 94M370 [33].

Plants of each variety were grown in 16.5-cm-diameter-plastic pots containing Pro-Mix-Bx potting mix (Premier ProMix, Lansing, MI USA) and covered with fine screen mesh cages. Plants were grown and maintained at greenhouse conditions described previously [34,35]. Groups of 200 apterous adult aphids of each species were starved for 12h before infestation

and released onto pots of 30 plants of each of the three wheat cultivars. There were three replicate pots for each cultivar. At 24-, 48-, 72- and 96h post-infestation, 30–40 aphids were collected from each of the three replicate pots of plants of each of the three wheat cultivars. Aphid samples from the four time points were pooled within each of the three biological replicates collected for each of the three feeding treatments and stored in RNAlater (Qiagen, GmbH, Hilden, Germany) according to the manufacturer's recommendations.

## Plant response to feeding by *S. graminum* biotype I

The responses of *D. noxia* to the three wheat lines used in this study have been previously assessed [28]; however, the ability of *S. graminum* to establish on these lines are unknown and bioassays were conducted to determine assess virulence. Assays were conducted in Manhattan, KS, using three cylindrical (10 cm diam x 9 cm tall) pots of variety tested, with each pot containing three seeds each of Yumar wheat (containing the *Dn4 D. noxia* resistance gene); 94M370 (containing the *Dn7 D. noxia* resistance gene); the *D. noxia* susceptible variety Yuma (*Dn0*); or a *S. graminum* resistant control TAM110. After germination, seedlings were thinned to one per pot, and randomly placed and grown in a growth chamber (Percival Scientific, Perry, Iowa USA) at 26:18C$^o$ day / night and a photoperiod of 14:10 [L:D] h. When plants reached the two-leaf stage at 10 d post-planting, five aged-synchronized 3 d old greenbug nymphs reared on susceptible Jagger wheat plants were placed on each test plant. Infested plants were caged in 8.5 cm diam x 51 cm tall plastic cylindrical cages with two side openings (5 cm diam) and one top opening (8.5 cm diam) covered with mite-proof mesh to reduce humidity inside the cage. The base of each cage was pressed ~ 0.5 cm below the soil to hold it in place. Two un-infested two-leaf stage plants of each variety were caged similarly and served as controls for plant dry weight change. This protocol allowed measurement of both the antibiosis and tolerance categories of resistance to *S. graminum*.

Antibiosis was determined by counting the mean total number of aphids on plants of each wheat genotype. Tolerance was measured as the per cent mean proportional dry weight change in leaves of each genotype. Mean proportional per cent dry weight changes (% DWT) were calculated as: [(mean dry weight of uninfested plants–mean dry weight of infested plants/mean dry weight of uninfested plants) x 100] [36]. An additional measure of tolerance was made by calculation of a tolerance index [37], which removes the potential bias of aphid population differences in tolerance measurements. Mean plant tolerance indices were calculated as (% DWT/total # aphids).

Reaction of plants of the three wheat genotypes to *S. graminum* feeding was assessed by measuring plant chlorosis, damage and dry weight; and the total number of *S. graminum* on each plant at 21 d post-infestation [38]. Plant chlorosis and damage scores were rated visually, using a scale of 1 = no chlorosis; 2 = >10% to 25% chlorosis; 3 = >25% to 50% chlorosis; 4 = >50% to 75% chlorosis; 5 = >76% to 100% chlorosis. After assessment, plants were cut at the soil level and placed on top of a piece of gridded cardstock (10 cm wide x 28 cm tall) coated with adhesive to trap aphids as plants dried at room temperature for 3 d. Dried plants were then removed from cards, bagged in aluminum pouches (11 cm wide x 12 cm tall) and placed in an oven (Precision, ThermoFisher Scientific, Waltham, MA USA) at 60˚C for 10 d. Dry weights were measured using a digital balance. The total number of aphids per sticky-card were counted using a stereoscope (Nikon SMZ645, Tokyo, Japan).

Total numbers of aphids and plant dry weight change data followed assumptions of normality and homogeneity of variances based on Kolmogorov-Smirnov, Levene, and Brown and Forsythe tests [39–41]. These data were analyzed using a normal distribution and PROC GLIMMIX [42], where plant variety was considered a fixed effect. Plant damage scores were

analyzed using an approximate normal distribution to estimate treatment differences between varieties. Plant chlorosis and plant tolerance index data did not follow assumptions of normality and homogeneity of variances. These data were analyzed using negative binomial distribution and Poisson distribution with log-link function, respectively, after verification of control of overdispersion with a Pearson Chi-square/DF test [43]. Degrees of freedom were estimated using the Kenward-Rogers method [44] when data failed to follow assumptions of normality and homogeneity of variances. When the F-test for type III effects was significant at $P < 0.05$, pairwise comparisons were conducted using Tukey's honestly significant difference at $\alpha = 0.05$ significance level [45].

### *D. noxia* and *S. graminum* RNA isolation, library preparation, and sequencing

Total RNA was isolated from the three biological replications collected from each of the three feeding treatments using the RNeasy Plus Kit (Qiagen, GmbH, Hilden, Germany) and treated with DNase. RNA was quality-checked using three different methods, including absorbance at 230, 260 and 280 nm on a NanoDrop spectrophotometer (Thermoscientific, Wilmington, DE USA), 1% agarose (RNase-free grade) gel electrophoresis using GelGreen staining (Biotium Inc., Hayward, CA USA) and by capillary electrophoresis using an RNA Nano Lab-Chip (Agilent, Santa Clara, CA USA) and an Agilent 2100 Bioanalyzer system. Overall, RNA was collected from 18 different samples (two aphid species x three feeding treatments x three biological replicates). These included *D. noxia* and *S. graminum* each fed plants of wheat genotypes that contained either the *Dn0*, *Dn4*, or *Dn7* genes. Approximately 1 μg of total RNA (100ng/μl) from each sample was used for library preparation with the Illumina TruSeq RNA sample preparation kit (Illumina Inc., San Diego, CA USA) per the manufacturer's recommendations. These libraries were validated, and a portion of each was diluted to a 10 nM concentration. Samples were separately barcoded for multiplexing, and libraries from all 18 samples were combined into two pools (each loaded in 1 lane) for a total of nine libraries per pool. A 1 x 100 bp single-end sequencing run was performed using an Illumina TruSeq single-read clustering Kit v3 and Illumina TruSeq SBS-HS v3 sequencing chemistry on an Illumina Hiseq 2500 sequencer. Library preparation and sequencing were conducted at the University of Kansas Medical Center, Kansas City, KS USA. Raw sequencing reads from *D. noxia* and *S. graminum* have been deposited in NCBI's Sequence Read Archive (SRA) under Bioproject PRJNA306025. SRA experiments SRX1494436 to SRX1494443 and SRX1494451 are derived from *S. graminum* and SRX1494444 to SRX1494451, SRX1494434, and SRX1494435 are derived from *D. noxia*.

### Primary sequence processing

Low quality sequences with mean quality scores <25 (min_qual_mean 25, trim_qual_type mean, Trim_qual_rule lt), reads consisting of more than 1% ambiguous bases (ns_max_p 1), and exact duplicates (derep 1), were removed using PRINSEQ [46] prior to transcriptome assembly for *S. graminum* and prior to read mapping for *D. noxia*. Further, low quality bases with PHRED scores <20 (trim_qual_left 20, trim_qual_right 20, trim_qual_window 2, trim_qual_step1), Illumina sequencing adapters, polyA/T tails (lc_method entropy and lc_threshold 70), and poly N tails containing five or more ambiguous bases (trim_ns_left 5 and trim_ns_left 5) were stripped from the reads. Reads shorter than 35 nt after quality trimming were also discarded. FastQC was used to validate the improved quality of the reads after quality filtering (https://www.bioinformatics.babraham.ac.uk/projects/fastqc/)

## *Schizaphis graminum* transcriptome assembly and abundance estimation

Quality filtered reads from all nine *S. graminum* samples were pooled and a *de novo* transcriptome assembly was performed using Trinity v.2.3.2 (50) with a kmer length of 25 (default), minimum contig length of 200 nt (default) and *in silico* normalization. After assembly, reads were pooled from each of the nine samples and mapped back to the transcriptome assembly using the align_and_estimate_ abundance.pl script with the RSEM method for abundance estimation [47] and Bowtie for read mapping [48]. Transcripts with <0.5 transcripts per million mapped reads (TPM) or transcripts representing < 10% of the expression value of the dominant isoform for each unigene, were removed from the transcriptome assembly. Protein coding regions of at least 100 amino acids in length were then identified using Transdecoder v.3.0.1 (https://transdecoder.github.io/) and the single highest scoring ORF for each transcript was retained using the single_best_orf option. Finally, transcripts containing no open reading frames were removed from the assembly. Functional annotations were then predicted for protein coding transcripts using Trinotate v.3.0.2 (https://trinotate.github.io/), which incorporated results from blastp/x (ncbi-blast v.2.6.0+) searches against the Swiss-Prot database (version 32 as of March 10, 2017), hmmer searches against the PFAM-A database, signalP searches, and TMHMM searches.

In addition, predicted ORFs were searched against the non-redundant protein database (downloaded on February 8, 2017) using blastp to identify any potential plant or bacterial transcripts in the assembly. In brief, the top five blastp matches with e-values ≤0.00001 were retained for each predicted coding region and taxonomic classifications were carried out using MEGAN's least common ancestor algorithm [49].

After removing non-coding transcripts, transcripts derived from microbes and 15,688 low abundance transcripts from the assembly as described above (which represented approximately 18% of the total number of assembled transcripts), reads from the nine libraries were re-aligned to the filtered transcriptome assembly individually using the same methods described previously. RSEM counts from each of the nine samples were concatenated into a single count matrix for differential expression analysis, which was conducted using edgeR [50]. Only transcripts with counts per million (CPM) values greater than one in at least two samples were tested for differential expression. Read counts were normalized using trimmed mean of M-values (TMM) and variances were estimated using tagwise dispersions. Pairwise comparisons between all possible sample combinations were used to identify genes that were differentially expressed in at least one sample using Fisher's Exact test. Differential expression analysis was performed at the unigene level. For the purposes of this study, unigenes were defined as Trinity transcripts that shared significant sequence similarity (≥97%) but had different structures and likely represented transcript isoforms derived from the same gene or locus. Unigenes with False-Discovery-Rate (FDR) corrected p-values ≤0.05 were considered differentially expressed.

Gene ontology (GO) enrichments were performed using GoSeq [51] and k-means analysis [52] was performed to identify groups of aphid genes with similar expression patterns across the three plant gene treatments. For GoSeq, the entire list of genes with CPM> = 1 in at least two samples were used as a reference to determine enrichment and nodes containing less than five genes were excluded from the analysis to control false discovery rate. Enrichment was determined using the Wallenius approximation ('pwf') option and categories with Benjamini-Hochberg adjusted p-values <0.05 were considered enriched. Enriched terms were dereplicated using REViGO [53] using medium similarity (0.7) and SimRel for semantic similarity measure. For k-means, the number of clusters that best represented the dominant expression profiles in the dataset was selected using the 'factoextra' [54] and the 'NbClust' packages [55]

implemented in the R statistical environment (version 3.3.1) [56]. The elbow, silhouette and gap statistic methods, and the majority rule of the 'fviz_nbclus' function from the NbClust package were consulted to select the number of clusters for k-means. Finally, KEGG pathway analysis was performed to determine the impact of the different feeding treatments on unigenes assigned to various core metabolic pathways. In brief, protein coding sequences were assigned to pathways using the KAAS server [57] with blastp searches (single-directional best hit method) against a database consisting of annotated *A. pisum* and *D. noxia* enzymes. Impacts to differentially expressed unigenes were visualized using the KEGG pathway mapper tool available at https://www.genome.jp/kegg/mapper.html. All transcripts containing protein coding genes have been submitted to NCBI's Transcriptome Shotgun Assembly database (GIML00000000). The entire transcriptome assembly containing both coding and non-coding genes has been deposited in USDA's Ag Data Commons at https://data.nal.usda.gov/dataset/de-novo-transcriptome-assembly-schizaphis-gramium-biotype-i-feeding-wheat and an annotated assembly of the transcripts that code for proteins is available at NCBI's Transcriptome Shotgun Assembly (TSA) database under the accession GIML00000000.

### *Diuraphis noxia* transcriptome assembly and abundance estimation

Reads derived from *D. noxia* biotype 1 were quality filtered using the same approach as described above for *S. graminum* and mapped to the *D. noxia* biotype 2 genome assembly version WGS Accession JOTR00000000.1 [3] using Hisat2 v.2.0.5 [58]. The number of reads mapped to each locus were summed using the FeatureCounts command in the Subread v.1.5.1 package [59]. The annotation files are available at ftp://ftp.ncbi.nlm.nih.gov/genomes/all/GCF/001/186/385/GCF_001186385.1_Dnoxia_1.0/. Differentially expressed genes were identified using edgeR and GO terms, KEGG pathways, and k-means analyses were performed using the same protocols as described above for *S. graminum*.

## Results

### Insect identities

Sequencing results of aphid amplification products confirmed that *D. noxia* U. S. biotype 1 samples were correctly identified and also revealed that the nine samples reported as *D. noxia* U. S. biotype 2 in Sinha et al. [2] were *Schizaphis graminum*, as indicated by trace chromatograph files (data not shown). The COI sequences from the transcriptome assembly also confirmed that the samples previously labeled *D. noxia* biotype 2 were *S. graminum* (99–100% identity). The identity of *S. graminum* biotype I was confirmed via a diagnostic assay using sorghum breeding line TX2783, which is resistant to biotype I and susceptible to biotype E [60,61] (data not shown).

### Plant and aphid responses to aphid feeding on plants of three wheat lines containing either *Dn0*, *Dn4* or *Dn7*

**Antibiosis of *D. noxia* biotype 1 has been previously observed on *Dn7* and *Dn4* plants.** Previous studies demonstrated that leaf damage in wheat plants containing the *Dn4* gene or the or *Dn7* gene is significantly less than in susceptible control *Dn0* plants when infested by *D. noxia* biotype 1 [20,62,63,64]. The only study conducted to date on categories of resistance in *Dn4* plants found no evidence of tolerance to biotype 1 [65]. Antibiosis resistance via reduced reproduction of biotype 1 has been demonstrated in aphids fed either *Dn4* or *Dn7* plants compared to those fed *Dn0* plants [63,65,66].

**Table 1. Mean (lower, upper CI) % chlorosis, damage score, number of *S. graminum* biotype I, % proportional dry weight change and tolerance index in plants containing *Dn4*- or *Dn7* genes, susceptible control *Dn0* plants and resistant control TAM110 plants infested by *S. graminum* at 21 d post infestation.**

| Genotype | Gene | Mean (lower, upper CI) | | | | |
| --- | --- | --- | --- | --- | --- | --- |
| | | % Chlorosis | Damage score | Total *S. graminum* | % DWT | TI |
| **Yuma** | *Dn0* | 76 (54.7, 105.6) a | 3.9 (3.2, 4.5) ab | 1089.6 (722.4, 1456.8) a | 48.9 (38.7, 59.0) a | 0.4 (0, 1.4) a |
| **Yumar** | *Dn4* | 87.5 (63.0, 121.4) a | 4.5 (3.8, 5.1) a | 569.5 (202.3, 936.7) a | 35.0 (24.8, 45.1) ab | 0.2 (0, 1.2) a |
| **94M370** | *Dn7* | 54 (38.7, 75.3) a | 3.2 (2.5, 3.8) b | 702.7 (335.5, 1069.9) a | 26.3 (16.1, 36.4) b | 0.5 (0, 1.5) a |
| **TAM110** | *Gb3* | 19 (13.4, 27.0) b | 1.8 (1.1, 2.4) C | 978.8 (611.6, 1346.0) a | 42.2 (32.0, 52.4) ab | 0.04 (0,1.0)a |

Means in each column followed by a different letter differ significantly based on a Tukey's HSD-mean separation test ($P < 0.05$).

DWT = Proportional dry weight change [(mean weight uninfested plant–weight infested plant)/mean weight uninfested plant) x 100]

TI = Tolerance index (DWT/Total # *S. graminum*)

**Plants of breeding line 94M370 exhibit resistance to *S. graminum* biotype I.** Mean percent plant chlorosis ($F_{3, 36} = 16.45$, $P < 0.0001$) and mean plant damage scores ($F_{3, 36} = 3.76$, $P = 0.02$) were significantly lower on TAM110 resistant control plants than on plants containing *Dn0*, *Dn4* or *Dn7* (Table 1). The mean damage of 94M370 plants was also significantly lower than that of *Dn4* plants and mean percent dry weight change (% DWT) in 94M370 plants was significantly less than that in *Dn0* plants ($F_{3, 36} = 12.34$, $P < 0.0001$), but there were no other significant differences in % DWT between varieties. There were no significant differences in total aphid number between varieties ($F_{3, 36} = 1.76$, $P = 0.17$). As a result, there were no significant differences between the tolerance index (% DWT/ # aphids) values of any varieties ($F_{3, 36} = 0.28$, $P = 0.83$). Overall, 94M370 plants were more resistant to *S. graminum* biotype I than plants containing *Dn0* or *Dn4* as shown by significantly decreased foliar damage (Table 1). 94M370 plants appear to be resistant to both *D. noxia* biotype 1 and *S. graminum* biotype I, there are differences in the mechanisms of resistance to each aphid. Foliar damage is significantly reduced in 94M370 plants in response to feeding by each species, but the reduction in *S. graminum* biotype I populations does not differ significantly at ($P < 0.05$), while numbers of *D. noxia* biotype 1 on 94M370 plants are significantly less than those on *Dn0* plants.

## *S. graminum* assembly metrics

The final *S. graminum* transcriptome assembly contained 23,982 transcripts derived from 13,534 protein coding unigenes with an average number of isoforms per unigene of 1.8 and an average GC content of 38.48%. Median transcript length was 1,032 nt and the total assembly length was 33.02 Mb (Table 2). When including only the single longest isoform per unigene, the median transcript length dropped to 942 nt and the total assembly size was reduced to 17.54 Mb. In addition, approximately 75% of the transcripts containing ORFs (18,075 transcripts), representing 71% of the protein coding unigenes (9,641 unigenes), had a significant blastp match to at least one Swiss-Prot protein (Table 2). Of the protein coding transcripts lacking BLASTP matches to Swiss-Prot, 1.4% (331 transcripts derived from 246 unigenes) had a significant BLASTX match to Swiss-Prot and 5.4% contained at least one PFAM domain (1,299 transcripts derived from 954 unigenes). Approximately 500 unigenes derived from *Buchnera* (2.5%) were also detected in the assembly, with a N50 contig size of 945 bp, total assembly length of approximately 401 kb, and an estimated GC content of 27.82%. In addition, 0.15% of the unigenes in the assembly were derived from plants, 0.38% were derived from other bacteria, and 0.05% were derived from fungi. Similar to the metabolic capacity of other *Buchnera* sp., unigenes coding for the metabolic pathways detected in this transcriptome

**Table 2. Assembly statistics for *S. graminum* transcriptome assembly.**

| All Transcripts | |
|---|---|
| Total Trinity Transcripts* | 23,982 |
| Total Trinity Unigenes* | 13,534 |
| Contig N50 (bp) | 2,812 |
| Contig N30 (bp) | 3,939 |
| Contig N10 (bp) | 6,181 |
| Median Contig Length (bp) | 1,785 |
| Average Contig Length (bp) | 2,156 |
| Total Assembled Bases (bp) | 51,713,882 |
| GC Content (%) * | 35.81 |
| **Longest Isoform per Gene** | |
| Contig N50 (bp) | 2,676 |
| Contig N30 (bp) | 3,773 |
| Contig N10 (bp) | 6,118 |
| Median Contig Length (bp) | 1,597 |
| Average Contig Length (bp) | 1,962 |
| Total Assembled Bases (bp) | 26,551,485 |

*Values that are the same regardless of whether all transcripts or only the longest isoform per Trinity gene were included in the calculations.

assembly included genes coding for enzymes involved in folate biosynthesis (5 unigenes), phenylalanine, tyrosine, and tryptophan biosynthesis (10 unigenes), lysine biosynthesis (9 unigenes), valine, leucine, and isoleucine biosynthesis (4 unigenes), and arginine biosynthesis (6 unigenes). Unigenes derived from partial 16S and 23S rRNAs assigned to the genus *Buchnera* were also recovered, which showed 95% and 100% nucleotide identity to *Buchnera aphidicola* strains associated with *S. graminum*, respectively.

## 94M370 plants had a more pronounced impact on global gene expression in *S. graminum* than *Dn4* plants

An average of 84.1% of the reads derived from each *S. graminum* sample were successfully mapped back to the transcriptome assembly and no major differences in mapping metrics were detected among any of the three aphid-plant diet treatment groups. The three replicates of the 94M370 treatment were highly correlated with one another ($R^2 \geq 0.90$) based on global expression profiles and the *Dn4* treatment was more similar to the *Dn0* treatment than it was to the *Dn7* treatment (Figs 1 and 2). Overall, 2,063 *S. graminum* unigenes were differentially expressed in at least one treatment with an FDR corrected p-value of ≤0.05 (Fig 3).

Consistent with the resistance of 94M370 to *S. graminum*, the majority of the differentially expressed genes that were identified were either up- or down-regulated in insects feeding on 94M370 plants. Lending support to this observation, K-means analysis led to the identification of two major gene clusters that each contained unigenes with similar expression patterns across the three feeding treatments (Fig 4). Although various statistical methods detected the presence of from two to 10 clusters of co-expressed genes in the data, the most frequently predicted number of clusters among the different methods used to detect co-expression was two clusters. Cluster 1 contained 1,175 unigenes expressed at lower levels in *S. graminum* fed 94M370 plants relative to those fed *Dn4* and *Dn0* plants (Fig 4A and 4C), while cluster 2 contained 888 unigenes more highly expressed in *S. graminum* fed 94M370 plants than those

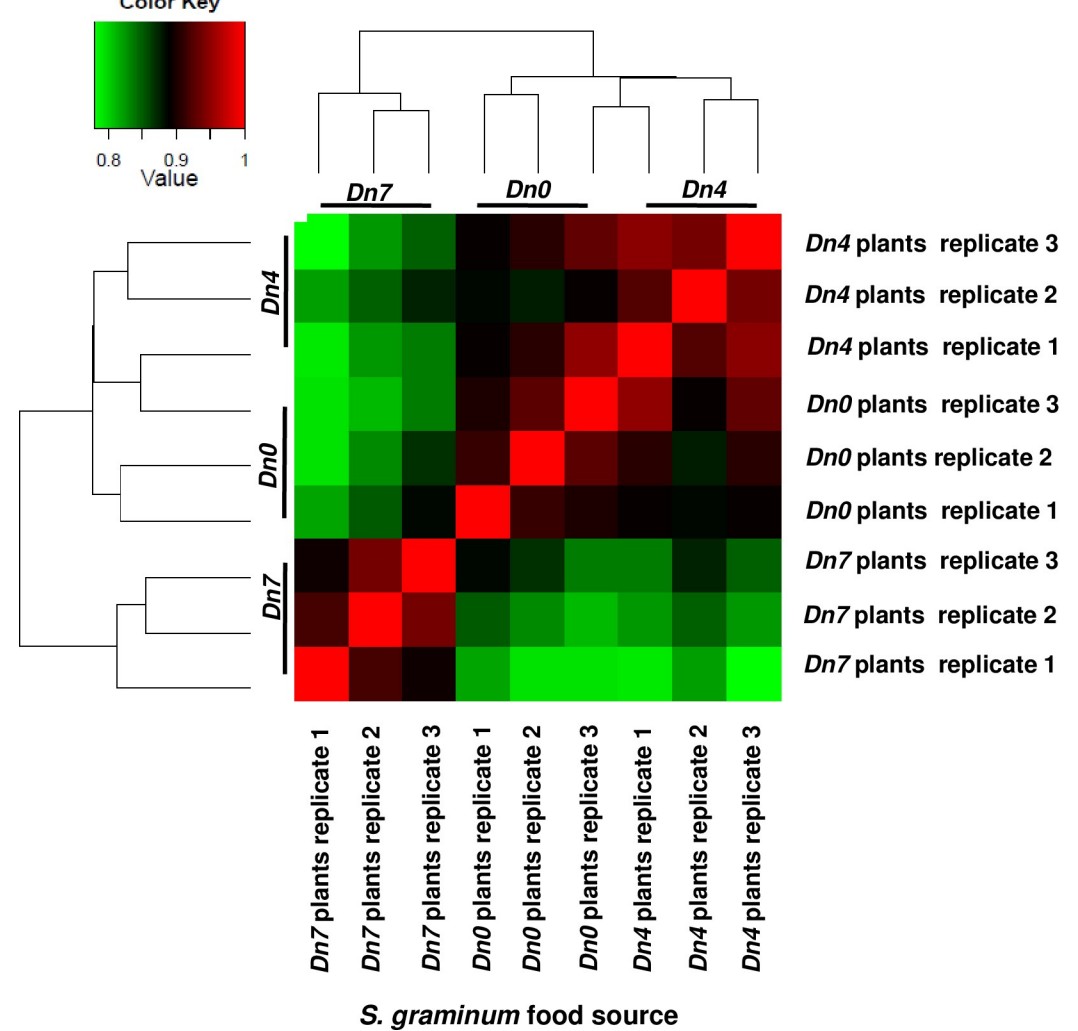

**Fig 1. EdgeR correlation matrix for DEGs expressed by greenbug,** *Schizaphis graminum*, **biotype I, fed plants containing no resistance genes (***Dn0***); the** *Diuraphis noxia* **biotype 1-resistant plants containing the** *Dn4* **resistance gene from wheat; or plants of 94M370 containing the** *Dn7* **gene from rye resistant to** *D. noxia* **and** *S. graminum*.

relative those fed *Dn4* and *Dn0* plants (Fig 4B and 4D). No other clusters of co-expressed unigenes could be identified through k-means, indicating that 94M370 plants containing the *Dn7* gene from rye had stronger effects on the global transcriptional profiles of *S. graminum* relative to those fed *Dn4* plants. However, a closer inspection of the clusters led to the identification of 159 unigenes within cluster 1 that were more highly downregulated and had more substantial log fold changes in *S. graminum* fed 94M370 plants compared to the other unigenes assigned to that cluster (Fig 5A; average log-fold change (LFC) = -2); as well as 78 unigenes within cluster 2 that were more highly upregulated in *S. graminum* fed 94M370 plants compared to the other unigenes assigned to that cluster (Fig 5B; average LFC = 3).

## GO enrichments for signal transduction and nucleic acid metabolism in *S. graminum* were associated with feeding on 94M370 plants

Of the cluster 1 unigenes in Fig 4, 284 were exclusively downregulated in *S. graminum* fed 94M370 plants relative to those fed *Dn0* plants, 862 were exclusively downregulated in *S.*

**Upregulated after feeding on *Dn7* plants relative to *Dn0* and *Dn4* plants**

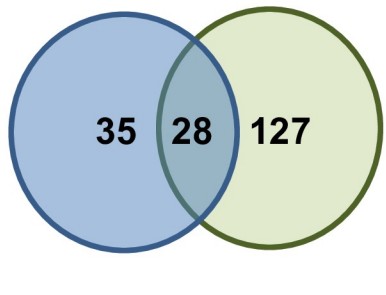

Dn0        Dn4

**Downregulated after feeding on *Dn7* plants relative to *Dn0* and *Dn4* plants**

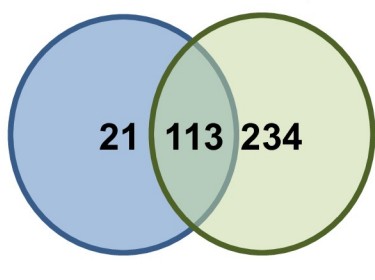

Dn0        Dn4

**Fig 2. *S. graminum* DEGs associated with feeding on wheat plants containing the *Dn7* resistance gene from 94M370 plants relative to plants containing the *Dn4* resistance gene or susceptible *Dn0* plants.**

*graminum* fed 94M370 plants relative to those fed *Dn4* plants and 29 were downregulated in *S. graminum* fed 94M370 plants relative to those fed *Dn4* and *Dn0* plants (Fig 4C). Of the cluster 2 unigenes, 46 were exclusively upregulated in *S. graminum* fed 94M370 plants relative to those fed *Dn0* plants, 648 were exclusively upregulated relative to those fed *Dn4* plants, and 194 were upregulated relative to those fed *Dn4* and *Dn0* plants (Fig 4D).

Overall, unigenes with lower expression levels in *S. graminum* fed 94M370 plants relative to those fed *Dn4* and *Dn0* plants (cluster 1) were enriched for GO categories linked to regulation of developmental process, cell migration, regulation of nucleic acid specific binding, cholesterol transporter activity, receptor activity, and signal transduction. Specific enriched terms included signal transduction activity (GO:0004871), signaling receptor activity (GO:0038023), oxidoreductase, acting on diphenols (GO:0016882), steroid transporting ATPase activity (GO:0034041), and regulatory region nucleic acid binding (GO:0001067) (Table 3). Other enriched terms included anatomical structure development (GO:004886), regulation of cell development (GO:0060284), regulation of neurogenesis (GO:0050767), morphogenesis of an epithelium (GO:0048513), regulation of response to stimulus (GO:0048585), biological adhesion (GO:0022610), axon guidance (GO:0042659), intrinsic component of membrane (GO:0031226), and negative regulation of serine/threonine kinase activity (GO:0071901) (S1 Table).

The upregulated unigenes in cluster 2 from *S. graminum* fed 94M370 plants were enriched for GO terms linked to nucleic acid metabolic process (GO:0090304), cellular nitrogen compound metabolic process (GO:0034641), double-strand break repair via homologous recombination (GO:0000724), DNA replication (GO:0006260), cell division (GO:0051301), ribosomal large subunit biogenesis (GO:0042273), and ribonucleoprotein complex biogenesis (GO:0022613) (Table 4). Other enriched GO terms included protein complex (GO:0043234), rRNA metabolic process (GO:0016072), Ada2/Gcn5/Ada3 transcription activator complex (GO:0005671), macromolecular complex (GO:0032991), ncRNA processing (GO:0034470), nucleolus (GO:0005730), and acetyltransferase complex (GO:1902493) (S2 Table).

Although not enriched, other GO categories that contained large numbers of unigenes impacted by feeding on 94M370 plants included cholesterol transport (GO:0030301; 9 of 26 annotated unigenes assigned to this category), melanin biosynthetic process (GO:0006469; 6 of 15 unigenes), and sensory organ development (GO:0007423; 37 of 169 unigenes), all of which were associated with differentially expressed genes (DEGs) (Table 5). Additionally, unigenes

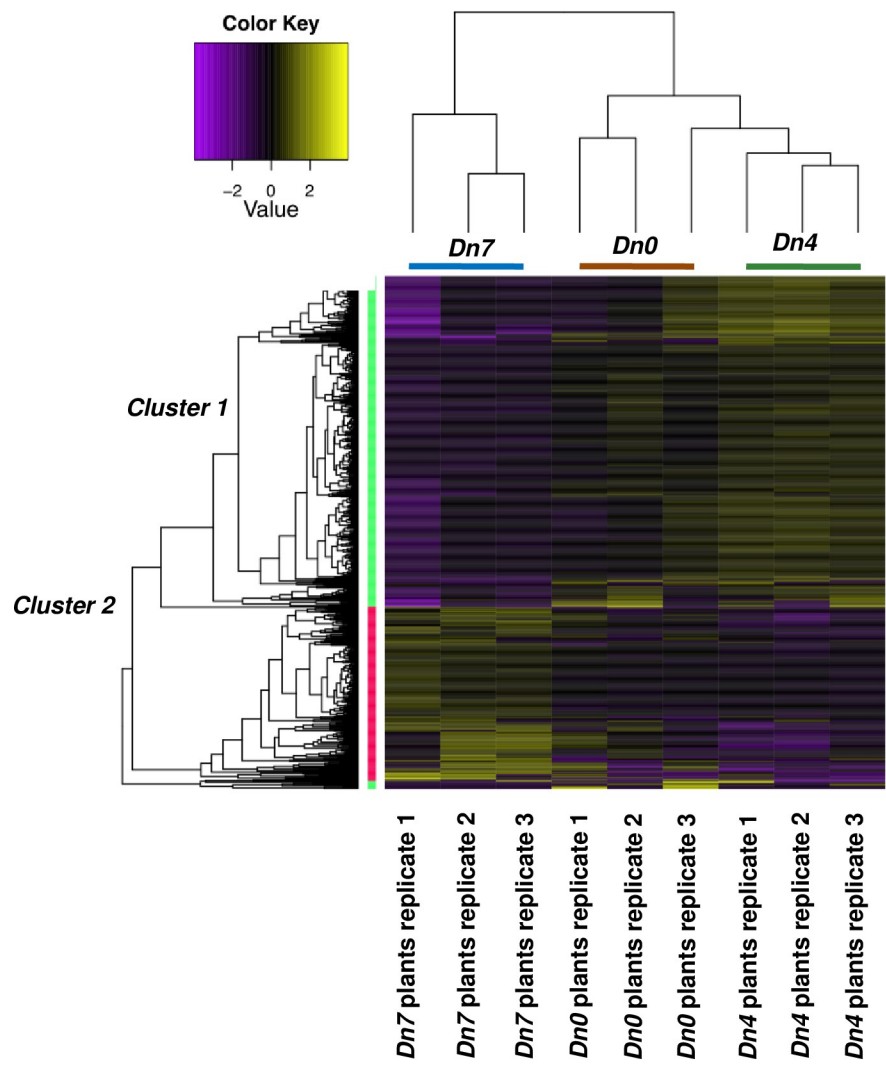

**Fig 3.** Clusters of co-expressed unigenes in *S. graminum* fed wheat plants containing the *Dn7* resistance gene from rye in 94M370 plants (blue bar); the *Dn4* resistance gene from wheat (green bar); or no resistance gene (*Dn0*) (orange bar).

coding for enzymes linked to mitotic cell cycle process (GO:1903047; 53 of 453 unigenes) and microtubule organization center (GO:0031023; 15 of 80 unigenes) were associated with DEGs upregulated in *S. graminum* fed 94M370 plants (Table 5).

## Expression of genes linked to actin cytoskeleton regulation and proteolysis were also impacted in *S. graminum* fed 94M370 plants

Pathway level analysis using KEGG assignments largely mirrored the results of the GO enrichments and identified additional impacts of the feeding treatments on gene expression in *S. graminum*. Cluster 1 downregulated unigenes were largely assigned to KEGG pathways for focal adhesion, several different signaling pathways, axon guidance, purine metabolism, actin cytoskeleton regulation, amino acid biosynthesis, lysine degradation, starch and sucrose metabolism, autophagy, apoptosis, endocytosis, protein digestion and absorption, glycerosphingolipid

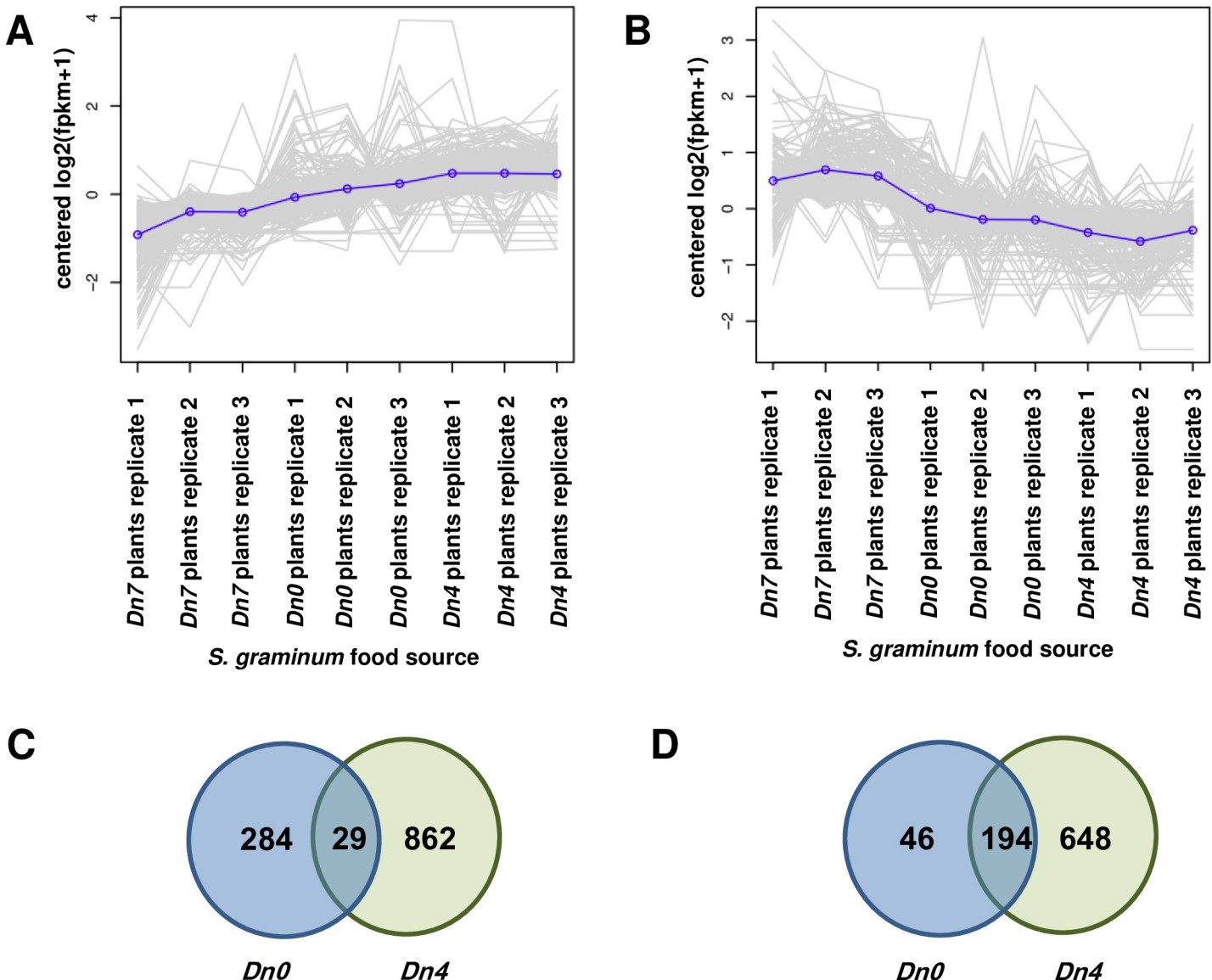

**Fig 4. K-means analysis of two major clusters of unigenes sharing common expression in *S. graminum* biotype I fed wheat plants containing the *Dn4 D. noxia* resistance or the *Dn7 D. noxia* resistance gene in 94M370 plants and susceptible *(Dn0)* plants.** A. Cluster 1–1,175 unigenes. B. Cluster 2–888 unigenes. C. Number of cluster 1 unigenes downregulated in *Dn7* relative to *Dn0* and *Dn4*. D. Number of cluster 2 unigenes upregulated in *Dn7* relative to *Dn0* and *Dn4*.

metabolism, cell cycle, galactose metabolism, lysosome, and melanogenesis (Table 6). Interestingly, in the amino acid biosynthesis pathway, unigenes coding for all enzymes linked to the synthesis of proline from glutamine, a unigene coding for cystathoine-γ lyase (cysteine biosynthesis), and serine/threonine ammonia-lyase (serine and/or threonine metabolism) were expressed at lower levels in *S. graminum* fed 94M370 plants compared to both plants containing *Dn0* or *Dn4* (Table 6).

Four unigenes coding for different portions of the lysine degradation pathway were also expressed at lower levels in *S. graminum* fed 94M370 plants while unigenes coding for enzymes linked to the degradation of sucrose and/or maltose (maltase-glucoamylase), gluconeogenesis (glycogen synthase), UDP-glucose metabolism (UDP-glucose pyrophoshorylase), and metabolism of chitin (chitinase) were also expressed at low levels in this treatment (Table 6).

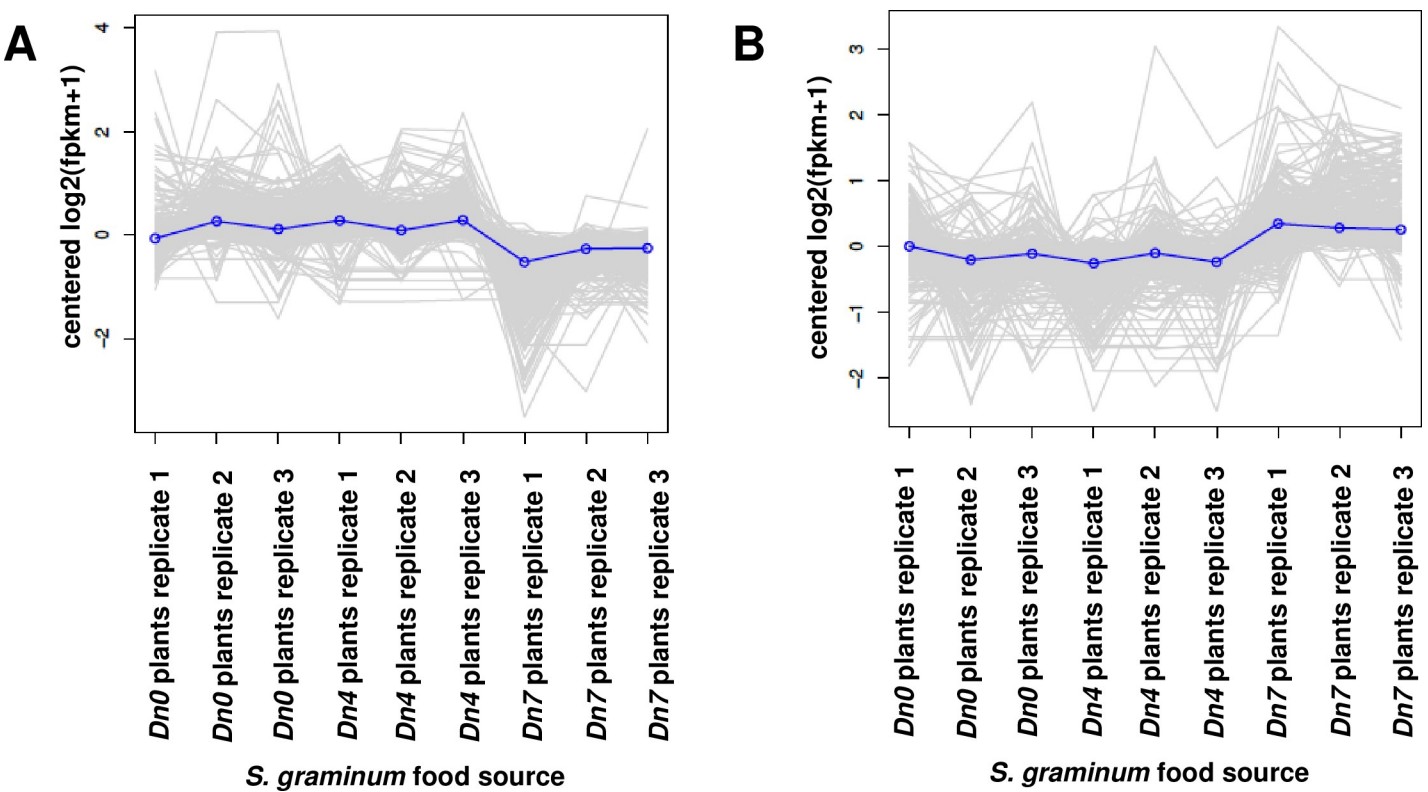

**Fig 5. K-means analysis of two major clusters of highly differentially expressed (2–8 fold) unigenes in *S. graminum* biotype I fed wheat plants containing the *D. noxia Dn7* resistance gene in 94M370 plants compared to other unigenes assigned to each cluster.** A. Cluster 1–159 highly downregulated unigenes. B. Cluster 2–78 highly upregulated unigenes.

Additionally, unigenes coding for four different glycosidases associated with lysosomal activity were also downregulated in *S. graminum* fed 94M370 plants compared to those fed plants containing *Dn0* or *Dn4* (Table 6). In contrast, upregulated unigenes associated with cluster 2 were assigned to pathways linked to ubiquitin mediated proteolysis, cell cycle, pyrimidine and purine metabolism, ribosome biogenesis, nucleotide excision repair, amino acyl tRNA biosynthesis, RNA degradation, endocytosis, and peroxisome (Table 7).

### Genes coding for detoxification enzymes were downregulated in in *S. graminum* fed 94M370 plants

Beyond these GO categories and KEGG pathways, *S. graminum* feeding on 94M370 plants expressed unigenes coding for proteins broadly linked to detoxification, digestion and growth or development (S3 and S4 Tables). Most of these unigenes were actually downregulated when compared to *S. graminum* fed plants containing either *Dn4* or *Dn0* (S3 Table). Those most strongly downregulated included a detoxification acyl transferase homologous to *nose resistant to fluoxetine 6 protein* (-8.6 log-fold change (LFC)), platelet-activating factor acetylhydrolase IB (-5.3 LFC), and a WD domain, G-beta repeat protein involved in tRNA binding (-6.5 LFC). Several developmental proteins were highly downregulated, including platelet-activating factor acetylhydrolase IB (-5.3 LFC), Dscam2 cell adhesion molecule-like protein (-7.4 LFC), a GCM motif protein (-6.2 LFC), and a DHHC palmitoyltransferase (-4.2 LFC) (S3 Table). A unigene coding for phosphate acetyltransferase involved in the metabolism of pyruvic acid was also highly downregulated (-6.6 LFC).

**Table 3. Enriched gene ontology (GO) molecular function terms for cluster 1 unigenes expressed at lower levels in *S. graminum* fed 94M370 plants compared to those fed either *Dn0* or *Dn4* plants.**

| Category | p-value* | # unigenes in category | | GO Molecular function term |
| | | DEs ** | All | |
|---|---|---|---|---|
| GO:0001067 | 0.000915 | 59 | 302 | Regulatory region nucleic acid binding |
| GO:0001071 | 0.017565 | 77 | 549 | Nucleic acid binding transcription factor activity |
| GO:0001067 | 0.000915 | 59 | 302 | Regulatory region nucleic acid binding |
| GO:0001071 | 0.017565 | 77 | 549 | Nucleic acid binding transcription factor activity |
| GO:0004871 | 0.044474 | 69 | 382 | Signal transducer activity |
| GO:0004872 | 0.000779 | 77 | 348 | Receptor activity |
| GO:0038023 | 0.024331 | 55 | 287 | Signaling receptor activity |
| GO:0004879 | 0.008224 | 7 | 12 | RNA polymerase II transcription factor activity, DNA binding |
| GO:0016682 | 0.044140 | 5 | 10 | Oxidoreductase activity on diphenols & related donors, $O_2$ acceptor |
| GO:0017127 | 0.016783 | 8 | 16 | Cholesterol transporter activity |
| GO:0034041 | 0.027280 | 7 | 14 | Sterol-transporting ATPase activity |
| GO:0019904 | 0.001449 | 43 | 158 | Protein domain specific binding |
| GO:0000977 | 0.015438 | 34 | 179 | RNA polymerase II regulatory region sequence- specific DNA binding |
| GO:0044212 | 0.000763 | 59 | 301 | Transcription regulatory region DNA binding |
| GO:0030215 | 0.009718 | 6 | 6 | Semaphorin receptor binding |
| GO:0005102 | 0.047627 | 49 | 245 | Receptor binding |
| GO:0042974 | 0.035408 | 4 | 5 | Retinoic acid receptor binding |
| GO:0003707 | 0.042884 | 8 | 21 | Steroid hormone receptor activity |

* Results dereplicated using REViGO and considered significant if false discovery rate corrected p-values were < 0.05. (Few cellular component and biological process function terms showed enrichment, S1 Table).

** DE = differentially expressed.

Four unigenes coding for glycoside hydrolase (GH) family 1 proteins were also downregulated in this comparison. All four GH family 1 unigenes have highest scoring blastp matches to myrosinases. Few GH family 1 genes have been functionally characterized in any aphid or insect species. However, many of these enzymes have known roles in digestive or detoxification processes in other organisms, such as degrading β-1,4-linked disaccharide sugars or metabolizing nitrogen- and sulfur-containing compounds. Unigenes coding for the most strongly upregulated proteins (S4 Table), included those for energy production (NADH-quinone oxidoreductase, 7.5 LFC); lipid homeostasis (glycerol-3-phosphate acyltransferase 1, 4.5 LFC); molting (zinc finger protein 862, 6.8 LFC); and DNA repair (DNA-directed RNA polymerase III subunit RPC5, 8.4 LFC). Several proteins related to DNA replication, gene expression, and chromatin structure were also highly upregulated. These included dUTP diphosphatase (7.9 LFC), leucine-tRNA ligase (7.2 LFC), and polycomb protein eed (6.2 LFC). Unigenes coding for proteins linked to growth/development and fatty acid biosynthesis and were also upregulated including protein inturned (6.4 LFC) and malonyl coA acyl carrier protein transacylase (6.9 LFC), respectively.

## *Dn4* plants had minimal impacts on gene expression in *S. graminum*

Although *Dn4* appeared to be susceptible to *S. graminum*, the expression levels of a few *S. graminum* unigenes were impacted by feeding on *Dn4* plants compared to *Dn0*, which included four up- and eight downregulated unigenes. Due to the low numbers of DEGs identified in *S. graminum* fed *Dn4*, no GO terms were enriched for any unigenes in this comparison (Fig 4). However, upregulated unigenes included a cytochrome P450 (CYP450; detoxification);

**Table 4. Enriched gene ontology (GO) biological process terms for cluster 2 unigenes expressed at higher levels in *S. graminum* fed 94M370 plants compared to those fed either *Dn0* or *Dn4* plants.**

| Category | p-value * | # unigenes in category | | GO Biological process term |
|---|---|---|---|---|
| | | DEs ** | All | |
| GO:0008152 | 0.001289 | 463 | 5663 | Metabolic process |
| GO:0009987 | 0.029541 | 552 | 7053 | Cellular process |
| GO:0034660 | 5.33E-11 | 72 | 391 | ncRNA Metabolic process |
| GO:0044085 | 1.80E-05 | 24 | 97 | Cellular component biogenesis |
| GO:0006807 | 1.82E-08 | 309 | 3203 | Nitrogen compound metabolic process |
| GO:0022402 | 0.001353 | 74 | 603 | Cell cycle process |
| GO:0071704 | 0.002858 | 433 | 5274 | Organic substance metabolic process |
| GO:1901360 | 1.71E-08 | 293 | 2996 | Organic cyclic compound metabolic process |
| GO:0044238 | 0.010144 | 410 | 5018 | Primary metabolic process |
| GO:0046483 | 3.78E-10 | 289 | 2840 | Heterocycle metabolic process |
| GO:0044237 | 2.03E-06 | 433 | 4998 | Cellular metabolic process |
| GO:0044270 | 0.011195 | 28 | 173 | Cellular nitrogen compound catabolic process |
| GO:0051301 | 0.009102 | 42 | 302 | Cell division |
| GO:0043170 | 6.36E-07 | 369 | 4078 | Macromolecule metabolic process |
| GO:0006725 | 1.69E-09 | 288 | 2879 | Cellular aromatic compound metabolic process |
| GO:0034641 | 1.13E-09 | 295 | 2954 | Cellular nitrogen compound metabolic process |
| GO:0044260 | 1.51E-09 | 354 | 3698 | Cellular macromolecule metabolic process |
| GO:0006281 | 2.41E-06 | 53 | 320 | DNA Repair |
| GO:0090305 | 2.00E-05 | 24 | 95 | Nucleic acid phosphodiester bond hydrolysis |
| GO:0090501 | 0.001289 | 12 | 33 | RNA Phosphodiester bond hydrolysis |
| GO:0016071 | 0.043868 | 46 | 368 | mRNA Metabolic process |
| GO:0006259 | 0.011636 | 83 | 790 | DNA Metabolic process |
| GO:0009451 | 0.007298 | 23 | 127 | RNA Modification |
| GO:0034470 | 1.70E-11 | 61 | 287 | ncRNA Processing |
| GO:0090304 | 3.71E-10 | 259 | 2455 | Nucleic acid metabolic process |
| GO:0006396 | 0.002822 | 54 | 400 | RNA Processing |
| GO:0033683 | 0.005565 | 7 | 13 | Nucleotide-excision repair, DNA Incision |
| GO:0006296 | 0.012274 | 5 | 7 | Nucleotide-excision repair, DNA Incision, 5'-to lesion |
| GO:0016070 | 6.29E-06 | 191 | 1845 | RNA Metabolic process |
| GO:0000469 | 0.000869 | 10 | 22 | Cleavage involved in rRNA processing |
| GO:0042273 | 0.004331 | 7 | 14 | Ribosomal large subunit biogenesis |
| GO:0006139 | 5.78E-10 | 278 | 2718 | Nucleobase-containing compound metabolic process |
| GO:1902589 | 0.017262 | 108 | 1036 | Single-organism organelle organization |
| GO:0022613 | 5.85E-05 | 19 | 69 | Ribonucleoprotein complex biogenesis |
| GO:0000726 | 0.039812 | 9 | 32 | Non-recombinational repair |
| GO:0006260 | 0.032719 | 24 | 151 | DNA Replication |
| GO:0000724 | 0.027628 | 12 | 48 | Double-strand repair via homologous recombination |
| GO:0006996 | 0.020083 | 135 | 1353 | Organelle organization |
| GO:1901361 | 0.038385 | 28 | 189 | Organic cyclic compound catabolic process |

* Results dereplicated using REViGO and considered significant if false discovery rate corrected p-values were < 0.05. (Few cellular component and molecular process function terms showed enrichment, S2 Table).

** DE = differentially expressed.

glutamate dehydrogenase (an important branch point between carbon and nitrogen metabolism) and a zinc finger protein (transcriptional regulation) (S5 Table), while downregulated

**Table 5. Additional GO terms associated with unigenes differentially expressed in S. *graminum*.**

| Category | p-value* | # unigenes in category | | Ontology | GO Term |
|---|---|---|---|---|---|
| | | DEs ** | All | | |
| **Cluster 1** | | | | | |
| GO:0030301 | 0.051804 | 9 | 23 | BP | Cholesterol transport |
| GO:0006469 | 0.051804 | 18 | 62 | BP | Negative regulation of protein kinase activity |
| GO:0042438 | 0.052377 | 6 | 15 | BP | Melanin biosynthetic process |
| GO:0007420 | 0.052377 | 21 | 87 | BP | Brain development |
| GO:0010605 | 0.053194 | 122 | 815 | BP | Negative regulation of macromolecule metabolic process |
| GO:0005319 | 0.056375 | 17 | 52 | MF | Lipid transporter activity |
| GO:0009890 | 0.058147 | 83 | 539 | BP | Negative regulation of biosynthetic process |
| GO:0048598 | 0.062139 | 45 | 226 | BP | Embryonic morphogenesis |
| GO:0000122 | 0.062482 | 47 | 276 | BP | Negative regulation RNA polymerase II promoter |
| GO:0046189 | 0.062717 | 8 | 24 | BP | Phenol-containing compound biosynthetic process |
| GO:0007423 | 0.066230 | 37 | 169 | BP | Sensory organ development |
| GO:0000987 | 0.066552 | 19 | 88 | MF | Core promoter proximal region sequence DNA binding |
| GO:0060429 | 0.066866 | 36 | 161 | BP | Epithelium development |
| **Cluster 2** | | | | | |
| GO:1903047 | 0.051324 | 53 | 453 | BP | Mitotic cell cycle process |
| GO:0044452 | 0.052332 | 11 | 46 | CC | Nucleolar part |
| GO:0006388 | 0.053082 | 5 | 10 | BP | tRNA Splicing, via endonucleolytic cleavage and ligation |
| GO:0006397 | 0.054059 | 40 | 310 | BP | mRNA Processing |
| GO:0000280 | 0.054343 | 31 | 224 | BP | Nuclear division |
| GO:0044451 | 0.056285 | 56 | 483 | CC | Nucleoplasm part |
| GO:0030687 | 0.056285 | 7 | 22 | CC | Preribosome, large subunit precursor |
| GO:0006308 | 0.056380 | 8 | 27 | BP | DNA Catabolic process |
| GO:0031125 | 0.056380 | 5 | 10 | BP | rRNA 3'-End processing |
| GO:0000110 | 0.069689 | 3 | 3 | CC | Nucleotide-excision repair factor 1 complex |
| GO:0031023 | 0.069689 | 15 | 80 | BP | Microtubule organizing center organization |

* Results dereplicated using REViGO and considered significant if false discovery rate corrected p-values were < 0.05.

** DE = differentially expressed.

unigenes included one gene each for coding for GDSL lipase (neurotransmitter), acyltransferase (detoxification), RNA-dependent DNA polymerase (transposase) major facilitator transporter (transmembrane transport), and a cleavage stimulation factor (polyadenylation) (S6 Table). The rest were hypothetical proteins that could not be annotated. In addition, four of the eight down-regulated DEGs in *S. graminum* fed *Dn4* plants were also downregulated in aphids fed 94M370 plants compared to those fed *Dn0* plants (S3 and S6 Tables), while only one, a zinc-finger protein, was upregulated in *S. graminum* fed both *Dn4* and 94M370 plants compared to those fed *Dn0* plants (S4 and S5 Tables). The four commonly downregulated unigenes coded for an acyltransferase, a lipase, a transporter, and a hypothetical protein.

## 94M370 plants had the most pronounced impacts on DEGs produced in *D. noxia*

Greater than 85% of the RNA-Seq reads derived from *D. noxia* biotype 1 successfully mapped to the biotype 2 reference genome, indicating the biotype 2 genome assembly to be a suitable reference for read mapping. No major differences in mapping metrics were detected among any of the three *D. noxia* treatment groups (feeding on *Dn0*, *Dn4*, or 94M370 plants) and the

**Table 6. KEGG pathway assignments for cluster 1 unigenes from *S. graminum*.**

| KEGG Pathway * | Number of unigenes ** |
|---|---|
| PI3K-Akt signaling | 17 |
| Insulin signaling | 17 |
| Focal adhesion | 17 |
| Axon guidance | 16 |
| AMPK signaling | 15 |
| MAPK signaling–fly | 14 |
| Hippo signaling–fly | 13 |
| Axon regeneration | 13 |
| Tight junction | 11 |
| Glucagon signaling | 11 |
| FoxO signaling | 11 |
| Hippo signaling | 10 |
| Adherens junction | 10 |
| Longevity regulating | 10 |
| cGMP-PKG signaling | 9 |
| MAPK signaling | 9 |
| Phospholipase D signaling | 9 |
| Wnt signaling | 9 |
| Rap1 signaling | 9 |
| Regulation of actin cytoskeleton | 9 |
| Purine metabolism | 9 |
| Lysine degradation | 8 |
| Cellular senescence | 8 |
| Biosynthesis of amino acids | 8 |
| Sphingolipid signaling | 8 |
| Starch and sucrose metabolism | 8 |
| Carbon metabolism | 8 |
| Autophagy–animal | 8 |
| Calcium signaling | 7 |
| Endocytosis | 7 |
| mTOR signaling | 7 |
| Ras signaling | 7 |
| ECM-receptor interaction | 7 |
| Protein digestion and absorption | 7 |
| Apoptosis–fly | 7 |
| Neurotrophin signaling | 7 |
| Vascular smooth muscle contraction | 7 |
| Amino sugar and nucleotide sugar metabolism | 6 |
| JAK-STAT signaling | 6 |
| Glycerophospholipid metabolism | 6 |
| Phosphatidylinositol signaling system | 6 |
| Gap junction | 6 |
| Cholinergic synapse | 6 |
| Circadian entrainment | 5 |
| Notch signaling | 5 |
| Melanogenesis | 5 |
| Glycolysis / Gluconeogenesis | 5 |

(*Continued*)

**Table 6.** (Continued)

| KEGG Pathway [*] | Number of unigenes [**] |
|---|---|
| Cell cycle | 5 |
| Galactose metabolism | 5 |
| Lysosome | 5 |
| Glycerolipid metabolism | 5 |
| Cell adhesion molecules | 5 |
| Phototransduction–fly | 5 |
| Vitamin digestion and absorption | 5 |
| TGF-beta signaling | 5 |
| Glycine, serine and threonine metabolism | 5 |
| Pyruvate metabolism | 4 |
| Citrate cycle | 4 |
| Sphingolipid metabolism | 4 |
| Ubiquitin mediated proteolysis | 4 |
| Pentose and glucuronate interconversions | 4 |
| Fatty acid metabolism | 4 |
| Salivary secretion | 4 |
| Protein processing in endoplasmic reticulum | 4 |
| Alanine, aspartate and glutamate metabolism | 4 |
| mRNA surveillance | 4 |
| Apoptosis | 4 |
| Inositol phosphate metabolism | 4 |
| ABC transporters | 4 |

[*] Mapped using https://www.genome.jp/kegg/mapper.html.

[**] Numbers of unigenes assigned to each pathway (pathways with >4 unigenes not shown).

three biological replicates within each treatment were highly correlated with one another ($R^2$ ≥0.90) (Fig 6). The 420 differentially expressed *D. noxia* unigenes in at least one treatment with an FDR corrected p-value of ≤0.05 formed 6 clusters (Fig 7). As in *S. graminum*, the global gene expression profiles of *D. noxia* fed *Dn0* and *Dn4* plants were more similar to one another in comparison to those fed 94M370 plants. A total of 376 DEGs were present in *D. noxia* biotype 1 fed wheat plants containing at least one of the three different plant *Dn* genes compared to *D. noxia* fed only *Dn0* plants (Fig 8). Of these differentially expressed unigenes, 204 were exclusively upregulated in *D. noxia* fed 94M370 plants relative to those fed *Dn0* plants and 30 were exclusively upregulated in *D. noxia* fed *Dn4* plants relative to those fed *Dn0* plants (Fig 8A). Among downregulated DEGs, 138 were exclusively downregulated in *D. noxia* fed 94M370 plants relative to those fed *Dn0* plants and 4 were exclusively downregulated relative to those fed *Dn4* plants (Fig 8B). In contrast to the expression data in Fig 7, K-means analysis indicated the presence of one to 10 clusters of co-expressed genes in *D. noxia* fed the three plant diets. However, among the statistical methods used for k-means optimization, five was the most frequently predicted number of clusters, as shown in Fig 9.

## GO enrichments for fatty acid synthase and glucosidase activity were detected in *D. noxia* fed 94M370 plants

Among the five clusters shown in Fig 9, *D. noxia* fed 94M370 plants consistently upregulated 212 unigenes at levels higher than in those fed *Dn0* and *Dn4* plants (cluster 1–33 unigenes,

**Table 7. KEGG pathway assignments for cluster 2 unigenes from *S. graminum*.**

| KEGG Pathway [*] | Number of unigenes [**] |
|---|---|
| RNA transport | 16 |
| Ubiquitin mediated proteolysis | 13 |
| Spliceosome | 11 |
| Cell cycle | 10 |
| Nucleotide excision repair | 9 |
| Ribosome biogenesis in eukaryotes | 9 |
| Protein processing in endoplasmic reticulum | 8 |
| RNA degradation | 8 |
| Aminoacyl-tRNA biosynthesis | 7 |
| Base excision repair | 7 |
| Endocytosis | 7 |
| mRNA surveillance pathway | 76 |
| Ribosome | 6 |
| DNA replication | 6 |
| Homologous recombination | 6 |
| Peroxisome | 6 |
| Mismatch repair | 6 |
| Meiosis–yeast | 5 |
| Lysosome | 5 |
| Autophagy–animal | 4 |
| RNA polymerase | 4 |
| Terpenoid backbone biosynthesis | 4 |
| Purine metabolism | 4 |
| Proteasome | 4 |
| Non-homologous end-joining | 4 |
| mTOR signaling pathway | 4 |
| Oxidative phosphorylation | 4 |
| AMPK signaling pathway | 4 |
| SNARE interactions in vesicular transport | 4 |
| Pyrimidine metabolism | 4 |
| NOD-like receptor signaling pathway | 4 |
| Basal transcription factors | 4 |
| Cellular senescence | 4 |

[*] Mapped using https://www.genome.jp/kegg/mapper.html.

[**] Numbers of unigenes assigned to each pathway (pathways with >4 unigenes not shown).

cluster 2–179 unigenes) (Fig 9A and 9B). Cluster 3 contained 49 unigenes differentially expressed in *D. noxia* fed both 94M370 and *Dn4* plants, relative to those fed *Dn0* plants (Fig 9C). Additionally, 159 unigenes (cluster 4–110 unigenes, cluster 5–46 unigenes) were strongly downregulated in *D. noxia* fed 94M370 plants relative to those fed *Dn0* plants and *Dn4* plants (Fig 9D and 9E).

Due to the low numbers of unigenes assigned to each of the six clusters of co-expressed genes, few GO terms were enriched. However, unigenes that were downregulated in *D. noxia* fed 94M370 plants were enriched for fatty acid synthase (GO:0004312; cluster 4); glucosidase activity (GO:0015926; cluster 4); oxidoreductase activity, acting on CH-CH group of donors (GO:0016628; cluster 4); cell cycle process (GO:0022402; cluster 5); and microtubule binding

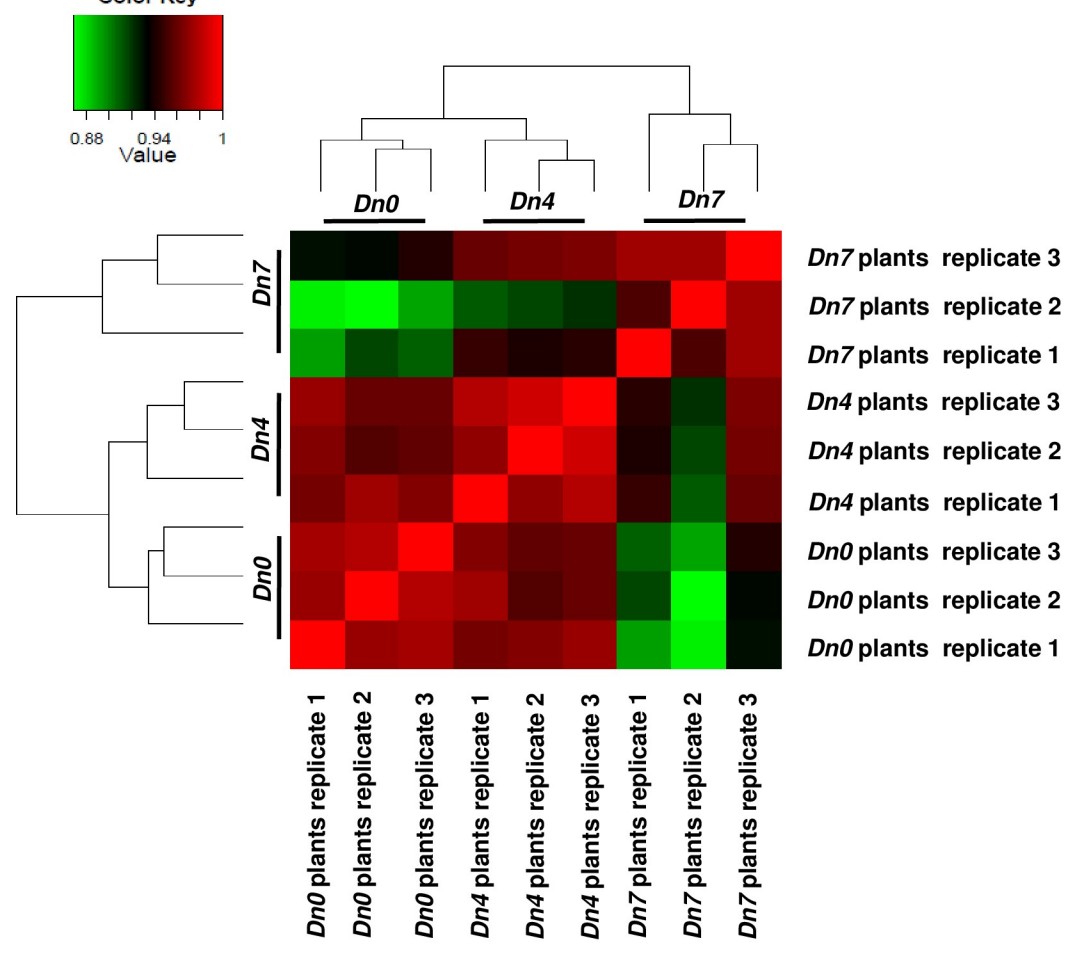

**Fig 6. EdgeR correlation matrix for DEGs expressed by Russian wheat aphid, *Diuraphis noxia*, biotype 1 fed plants containing no resistance genes (*Dn0*); the *D. noxia Dn4* resistance gene from wheat; or the *D. noxia Dn7* gene from rye.**

(GO:0008017; cluster 5) (Table 8). Unigenes that were upregulated in *D. noxia* fed 94M370 plants relative to *D. noxia* fed either *Dn4* plants or *Dn0* plants were enriched with GO terms linked to actin filament polymerization (GO:0030838; cluster 3) and protein K63-linked ubiquitination (GO:0070534; cluster 3). No other GO categories were enriched or highly abundant among the unigenes that were upregulated in *D. noxia* fed 94M370 plants (clusters 1 and 2) or cluster 6.

## Genes coding for lysosomal enzymes and purine metabolism were similarly impacted in *D. noxia* and *S. graminum* fed 94M370 plants

Additionally, KEGG pathway analysis identified unigenes associated with several metabolic pathways were impacted, particularly in *D. noxia* fed 94M370 plants relative to those fed either *Dn0* or *Dn4* plants. These included unigenes coding for enzymatic components of lysosomes and unigenes associated with thiamine absorption, autophagy, and signaling pathways, which were associated with upregulated unigenes (Table 9).

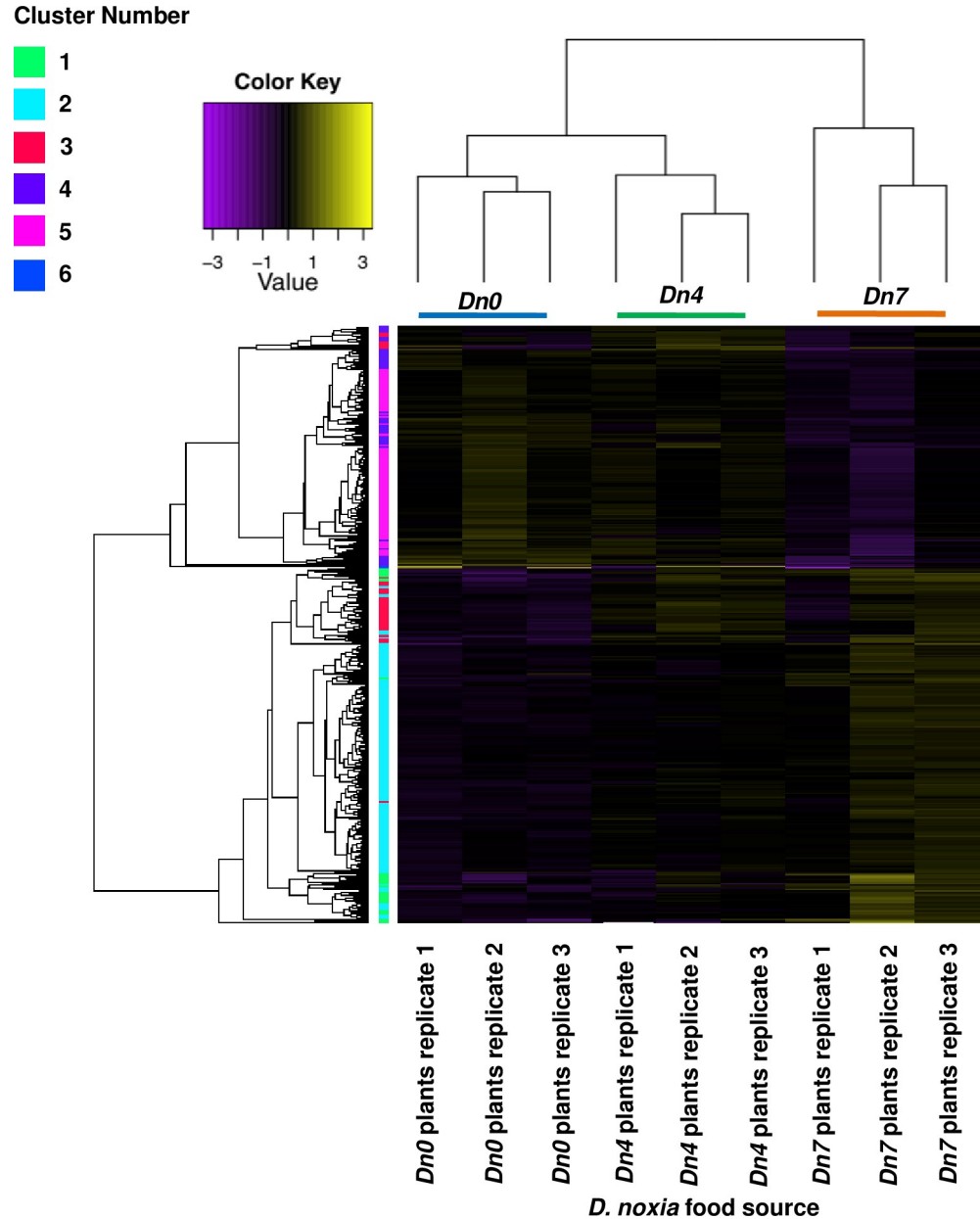

**Fig 7. Clusters of co-expressed unigenes in *D. noxia* biotype 1 fed wheat plants containing the *Dn7* resistance gene from rye (blue bar); the *Dn4* resistance gene (green bar); or no resistance gene (*Dn0*) (orange bar).**

In contrast, downregulated unigenes in this comparison were associated with purine metabolism (primarily inosine monophosphate biosynthesis and ribonucleotide reductase associated with the formation of deoxyribonucleotides from ribonucleotides). Genes linked to purine metabolism were also downregulated in *S. graminum* (Tables 6 and 7). Others unigenes downregulated in *D. noxia* in this comparison included those involved in protein processing in the endoplasmic reticulum, glutathione metabolism, and two enzymes associated with oxidative phosphorylation (Table 10). Due to the lower number of DEGs in the *Dn4* versus *Dn0* comparison, few metabolic pathways were impacted.

## A

**Upregulated after feeding on *Dn7* and/or *Dn4* plants relative to *Dn0***

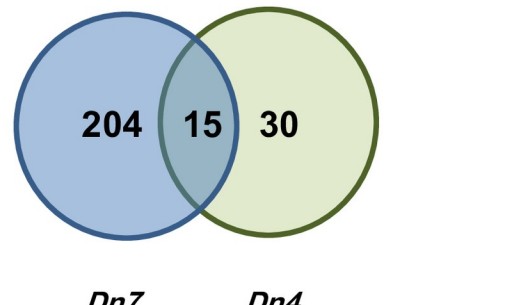

*Dn7*          *Dn4*

## B

**Downregulated after feeding on *Dn7* and/or *Dn4* plants relative to *Dn0***

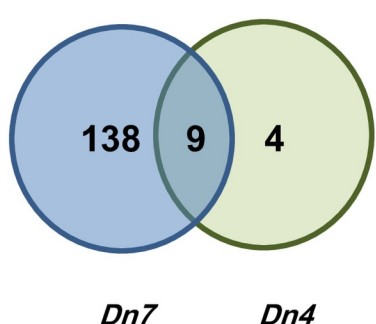

*Dn7*          *Dn4*

**Fig 8. *D. noxia* DEGs associated with feeding on wheat plants containing the *Dn7* or *Dn4* resistance genes relative to feeding on susceptible *Dn0* plants.**

### Expression levels of detoxification genes exhibited divergent expression patterns in *D. noxia* and *S. graminum* on 94M370 plants

*D. noxia* fed 94M370 plants upregulated between twice as many unigenes coding for proteins linked to detoxification and 5x more unigenes coding for proteins linked to nucleic acid processing or signaling than unigenes than *D. noxia* fed *Dn4* or *Dn0* plants (S7 Table). In contrast, *S. graminum* fed 94M370 plants downregulated most of the unigenes coding for proteins linked to detoxification (S3 Table). Unigenes associated with stress response and detoxification in *D. noxia* included two lipases, four CYP450 proteins, three phosophlipases, two glucose dehydrogenases, a multicopper oxidase, a cysteine proteinase inhibitor, two trypsins, a serpin, and a Kazal-type serine protease inhibitor were upregulated in this treatment (S7 Table). However, the fold change levels of all of these unigenes were much lower (0.75 LFC to 1.25 LF) than fold change levels of unigenes expressed by *S. graminum* (S4 Table). In addition, several unigenes linked to nutrient acquisition/transport and general metabolism were also upregulated, coding for proteins linked to transport of folates (2), sugar (2), acetyl-coA (2); fatty acid desaturase (1), gluconeogenesis, and vitamin A metabolism (1) (S7 Table).

Unigenes coding for detoxification and digestion were also predominant among the genes that were downregulated in *D. noxia* fed 94M370 plants relative to those fed *Dn0* plants, but again at lower expression levels than in *S. graminum*. Strongly downregulated detoxification unigenes included those coding for γ-glutamyltranspeptidase (-1.2 fold); heat shock protein (HSP) 70 (-2.2 and -0.35 fold); HSP90 (1); HSP60 (1); lipocalin (1); papain family cysteine protease (2); trypsin (1); ubiquitin (1), and UGT (3) (S8 Table). Additionally, unigenes coding for several key digestive enzymes and nutrient transporters were also downregulated in this treatment, including those coding for α-amylase (3), fatty acid synthase (3); lysosomal α-mannosidase (1); GH family 1 protein (1); reduced folate carrier (1); sugar (and other) transporters (3); and sulfate permease (1) (S8 Table). Unigenes coding to mitochondrial activities were also downregulated in *D. noxia* fed 94M370 plants, including those coding for cytochrome c oxidase proteins (1), cytochrome b561, as were unigenes associated with cell cycle activities, including cyclin (2); cell division protein (1); and Chromo (CHRomatin Organizaton MOdifer) domain protein (1 (S8 Table)). More unigenes coding for structural proteins were downregulated in *D. noxia* fed 94M370 plants compared to those that were upregulated (S7 and S8

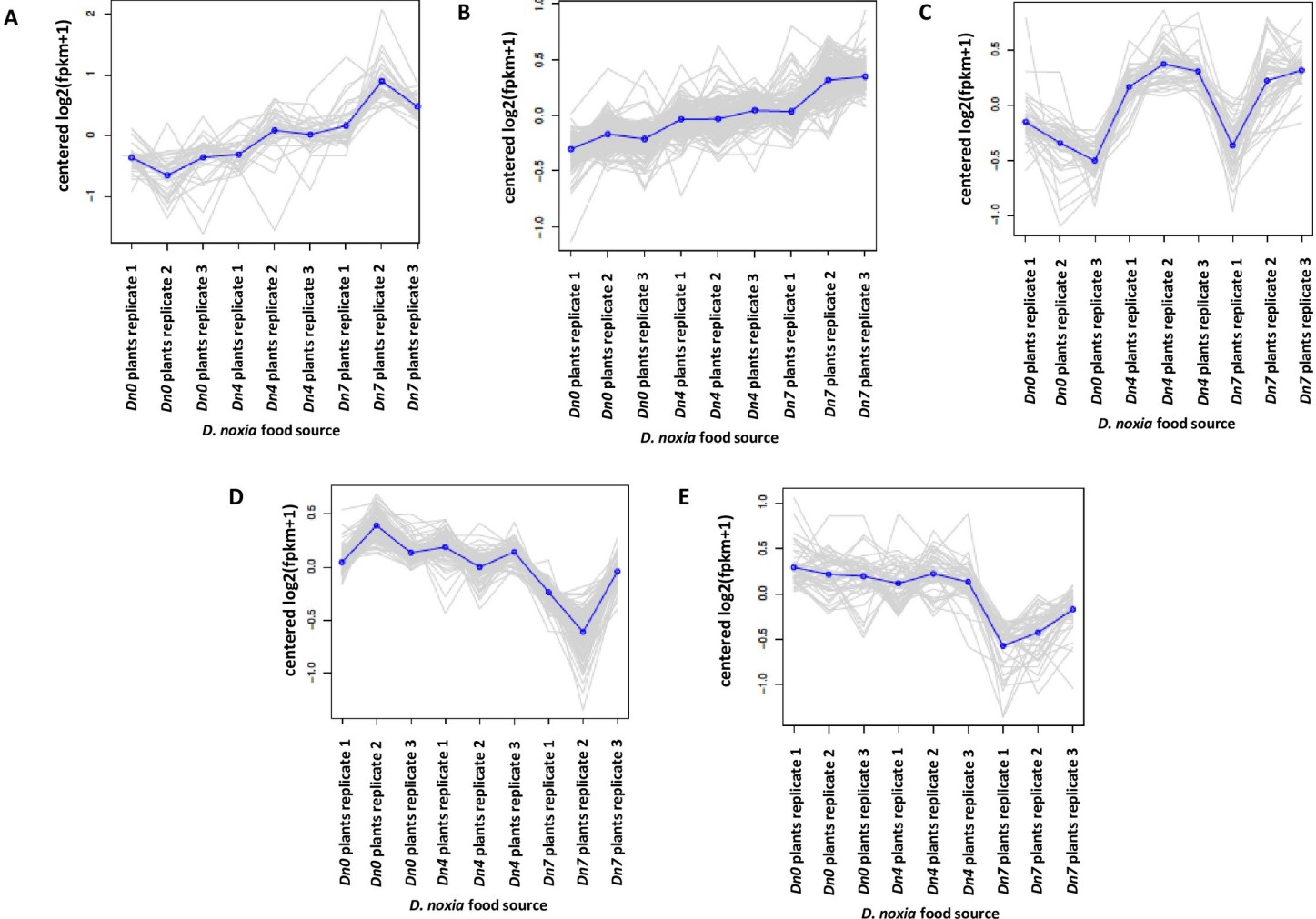

**Fig 9. K-means analysis of five major clusters of unigenes sharing common expression in *D. noxia* biotype 1 fed wheat plants containing the *Dn4* and *Dn7 D. noxia* resistance genes and susceptible (*Dn0*) plants.** A. Cluster 1–33 unigenes; B. Cluster 2–179 unigenes; C. Cluster 3–49 unigenes; D. Cluster 4–110 unigenes; E. Cluster 5–46 unigenes. Cluster 6 (2 unigenes) was omitted due to the low number of unigenes assigned to this group.

Tables). These unigenes were annotated as cadherin (1); insect cuticle protein (1); laminin (1); kinesin motor domain (1); and microtubule binding proteins (4).

## Impacts of *Dn4* plants had stronger effects on global gene expression in *D. noxia* than in *S. graminum*

Consistent with the antibiosis effects of *Dn4* plants on *D. noxia*, ingestion of phloem sap from *Dn4* plants had a much higher impact on unigene expression relative to *S. graminum*. Upregulated unigenes were predominantly linked to stress response and coded for ABC transporter (1); CYP450 (2); glucose dehydrogenase (1); lipase (1); multicopper oxidase (1); serpin protease inhibitor, and protein Spaetzle (1; ligand for Toll receptor) (S9 Table). The same unigenes coding for multicopper oxidase, lipase, and Spaetzle were also upregulated in *D. noxia* fed 94M370 plants when compared to those fed *Dn0* plants (S7 and S9 Tables). Several unigenes whose products were linked to growth and development were also upregulated, including two unigenes coding for peritrophin A proteins (2), one unigene coding for a GH 18 chitinase

**Table 8. Enriched gene ontology (GO) terms for unigenes differentially expressed in *D. noxia* fed 94M370 plants compared to those fed either *Dn0* or *Dn4* plants.**

| Category | Cluster # | p-value * | # unigenes in category | | Ontology *** | GO Term |
|---|---|---|---|---|---|---|
| | | | DEs ** | All | | |
| GO:0051125 | 3 | 0.002139 | 4 | 11 | BP | Regulation of actin nucleation |
| GO:0030904 | 3 | 0.002864 | 4 | 14 | CC | Retromer Complex |
| GO:0070534 | 3 | 0.004772 | 4 | 17 | BP | Protein K63-linked ubiquitination |
| GO:0030838 | 3 | 0.02096 | 4 | 29 | BP | Positive regulation of actin filament polymerization |
| GO:0004312 | 4 | 0.015731 | 4 | 27 | MF | Fatty acid synthase activity |
| GO:0016021 | 4 | 0.015731 | 21 | 1995 | CC | Integral Component of Membrane |
| GO:0015772 | 4 | 0.015731 | 4 | 40 | BP | Oligosaccharide Transport |
| GO:0016418 | 4 | 0.016816 | 3 | 16 | MF | S-Acetyltransferase Activity |
| GO:0072330 | 4 | 0.019319 | 5 | 87 | BP | Monocarboxylic Acid Biosynthetic Process |
| GO:0031224 | 4 | 0.02167 | 21 | 2063 | CC | Intrinsic Component of Membrane |
| GO:0031177 | 4 | 0.022696 | 3 | 18 | MF | Phosphopantetheine Binding |
| GO:0015926 | 4 | 0.023024 | 3 | 17 | MF | Glucosidase Activity |
| GO:0032787 | 4 | 0.023271 | 7 | 223 | BP | Monocarboxylic Acid Metabolic Process |
| GO:0019842 | 4 | 0.026193 | 4 | 51 | MF | Vitamin Binding |
| GO:0016746 | 4 | 0.035667 | 6 | 167 | MF | Transferase Activity, Transferring Acyl Groups |
| GO:0033218 | 4 | 0.035667 | 5 | 106 | MF | Amide Binding |
| GO:0016628 | 4 | 0.038374 | 3 | 22 | MF | Oxidoreductase Activity, Acting on CH-CH Donor Gp, NAD or NADP as Acceptor |
| GO:0051233 | 5 | 6.77E-05 | 6 | 11 | CC | Spindle Midzone |
| GO:0000910 | 5 | 0.001809 | 9 | 68 | BP | Cytokinesis |
| GO:0061640 | 5 | 0.002413 | 8 | 53 | BP | Cytoskeleton-Dependent Cytokinesis |
| GO:0022402 | 5 | 0.008538 | 20 | 449 | BP | Cell Cycle Process |
| GO:1903047 | 5 | 0.011333 | 16 | 306 | BP | Mitotic Cell Cycle Process |
| GO:0008017 | 5 | 0.022 | 9 | 90 | MF | Microtubule binding |
| GO:0030496 | 5 | 0.022 | 7 | 59 | CC | Midbody |
| GO:0032465 | 5 | 0.024387 | 5 | 23 | BP | Regulation of Cytokinesis |
| GO:0015631 | 5 | 0.049628 | 9 | 113 | MF | Tubulin Binding |
| GO:0006189 | 5 | 0.049628 | 3 | 6 | BP | 'De novo' IMP Biosynthetic Process |
| GO:1902850 | 5 | 0.049628 | 5 | 29 | BP | Mitosis Microtubule Cytoskeleton Organization |

* Results dereplicated using REViGO and considered significant if false discovery rate corrected p-values were < 0.05.

** DE = differentially expressed.

*** CC = cellular component; BP = biological process; MF = molecular function.

**Table 9. KEGG pathway assignments for genes upregulated in *D. noxia* fed 94M370 plants relative to *D. noxia* fed *Dn0* plants.**

| KEGG Pathway * | Number of unigenes ** |
|---|---|
| Biosynthesis of amino acids | 4 |
| Lysosome | 4 |
| Autophagy–animal | 3 |
| Vitamin digestion and absorption | 3 |
| AMPK signaling pathway | 3 |

* Mapped using https://www.genome.jp/kegg/mapper.html. *

** Numbers of unigenes assigned to each pathway (pathways with >3 unigenes not shown).

**Table 10. KEGG pathway assignments for genes downregulated in *D. noxia* fed 94M370 plants relative to *D. noxia* fed *Dn0* plants.**

| KEGG Pathway [*] | Number of unigenes [**] |
|---|---|
| Cell cycle | 6 |
| Protein processing in endoplasmic reticulum | 5 |
| RNA transport | 5 |
| Lysosome | 4 |
| Spliceosome | 4 |
| Fatty acid metabolism | 4 |
| Purine metabolism | 4 |
| Biosynthesis of unsaturated fatty acids | 3 |
| PPAR signaling pathway | 3 |
| Glutathione metabolism | 3 |
| Apoptosis | 3 |
| AMPK signaling pathway | 3 |

[*] Mapped using https://www.genome.jp/kegg/mapper.html.

[**] Numbers of unigenes assigned to each pathway (pathways with >3 unigenes not shown).

linked to chitin remodeling, one unigene coding for a JHBP, and three unigenes coding for insect cuticle proteins (S9 Table). Unigenes coding for reduced folate carriers (2; also upregulated in *D. noxia* fed 94M370 plants) and general odorant binding proteins (PBP/GOBP) were also upregulated (S9 Table). Interestingly, nine of the 13 unigenes that were downregulated in *D. noxia* fed *Dn4* plants were also downregulated in *D. noxia* fed 94M370 plants, which included unigenes coding for α-amylase (2), HSP70 (1); insect cuticle protein (1); papain family cysteine protease (1); and major facilitator superfamily transporters (2) (S8 and S10 Tables). The remaining four unigenes that were downregulated exclusively in *D. noxia* fed *Dn4* plants coded for hypothetical proteins (3) and a second insect cuticle protein (1) (S10 Table).

## Discussion

The molecular bases of arthropod virulence to plant arthropod resistance genes are poorly understood. However, evidence to date suggests the involvement of components from both the salivary glands and midgut that function in virulence of *D. noxia* and *S. graminum*. For example, [67] identified five major proteins from secreted saliva of virulent and avirulent *D. noxia* biotypes, however their function as virulence factors remains unproven. Similarly, Nicholson and Puterka [23] identified quantitative variation in the salivary proteomes of four differentially virulent *S. graminum* biotypes for glucose dehydrogenase, carbonic anhydrase, and an abnormal oocyte protein, yet their function as virulence factors also remains unproven. It is also interesting to note that Ji et al. [68] identified salivary gland secretory proteins that are differentially expressed in biotypes of a related Hemipteran species, the brown planthopper, *Nilaparvata lugens* (Stal), but again, their function as virulence factors remains unproven.

Prior to the *D. noxia* biotype 2 genome assembly, significant differences were shown to exist between the midgut transcriptomes of biotypes 1 and 2 after ingestion of phloem sap from wheat plants containing the *Dn4* gene [34]. The midgut of avirulent biotype 1 was shown to express many more protease inhibitors than that of virulent biotype 2, but many more proteases are expressed in the midgut of biotype 2 than biotype 1. These results suggested that the avirulent biotype produces protease inhibitors in response to plant proteases produced by biotype 1 resistant plants [69], and that virulent biotype 2 produce trypsin-like and

chymotrypsin-like serine protease counter-defenses to overcome biotype 1-resistant plants. In addition, biotype 1 fed *Dn4* plants upregulate serine proteinase inhibitors and downregulate cysteine proteinases [34]. In contrast, no proteinases or proteinase inhibitors were differentially expressed by *S. graminum* fed on wheat plants containing the *Dn4* gene, although the expression levels of other detoxification genes including CYP450 were upregulated. The differential expression of many more detoxification genes by *D. noxia* than by *S. graminum* suggests that ingestion of phloem sap from *Dn4* plants potentially has a greater impact on *D. noxia* biotype 1 physiology than that of *S. graminum* biotype I. This is also consistent with the antibiosis resistance displayed by *Dn4* plants. The high fitness costs associated with high expression of P450s by *D. noxia* biotype 1 for prolonged periods of time to deal with the antibiosis effects could lead to the reduced of fecundity and reproduction of *D. noxia* biotype 1 individuals [70].

Bansal et al. [11] compared transcriptomes of soybean aphid, *Aphis glycines*, fed plants containing the *Rag1 A. glycines* resistance gene, to those fed susceptible plants. Serine proteases are also up-regulated in *A. glycines* fed *Rag1* plants, while proteases are absent from genes down-regulated in *A. glycines*. Finally, a much larger subset of *Buchnera aphidicola* genes were expressed in biotype 2 than biotype 1, as well as a significantly higher expression of tRNALeu, strongly suggesting differences in titer levels of *B. aphidicola* and/or leucine metabolism may contribute to biotype 2 virulence. Significantly greater numbers of copies the *B. aphidicola leuA* gene have also been detected in biotype 2 than in biotype 1 [71]. In contrast, virtually nothing is known about the *S. graminum* transcriptomic responses to any type of wheat genes.

One of the keys to understanding the transcriptional responses of *D. noxia* and *S. graminum* to 94M370 plants is linked to the differences in responses of 94M370 plants to each aphid species. The phenotype of 94M370 is well-documented to both *D. noxia* biotypes 1 and 2, and plants possessing the 1BL.1RS translocation exhibit significantly reduced leaf chlorosis, leaf rolling, and *D. noxia* population development when compared to susceptible *Dn0* plants [62,63,72].

The cross resistance of 94M370 plants to *S. graminum* is characterized by reduced foliar damage and reduced plant dry weight loss in comparison to plants containing susceptible lines such as *Dn0* or *Dn4* (Table 1). However, the lack of a corresponding reduction in *S. graminum* population development is interesting. Reduced damage to arthropod herbivory is often indicative of tolerance [6]. Yet when plant dry weight changes were proportionalized for numbers of *S. graminum* produced, tolerance, as measured by the plant tolerance index, could not be demonstrated. These results suggest that 94M370 plants may possess a resistance mechanism unrelated to reduced plant tissue loss or reduced aphid population growth that contributes to protection of foliar tissues from *S. graminum*-related chlorosis. Although the responses of 94M370 plants to *S. graminum* appear be related to factors from rye in the *Dn7* chromosomal translocation, these responses may be linked to factors contributed by Gamtoos wheat used to create 94M370, or an interaction of factors in rye and Gamtos. Regardless, further studies are now necessary to map the genomic regions in 94M370 associated with *S. graminum* resistance.

Results of the current study indicate completely different functional themes in *S. graminum* transcriptomic responses to plants containing the *Dn4 D. noxia* resistance gene and to 94M370 plants containing the *Dn7 D. noxia* resistance gene when compared to those occurring in *D. noxia*. *S. graminum* is a member of the Macrosiphini aphid tribe and a generalist, feeding on barley, fescue, maize, oat, rice, sorghum and wheat. When challenged by plants containing the *Dn7* gene from rye, *S. graminum* generated two unigene clusters—an up-regulated cluster of ~880 unigenes and a down-regulated cluster of ~1,100 unigenes. When fed on either *Dn4* or *Dn7* plants, *S. graminum* down-regulated unigenes primarily involving nucleic acid binding, structural development, signal transduction and general metabolism (Table 3). Conversely, *S. graminum* fed plants *Dn4* or 94M370 plants *containing Dn7* up-regulate

unigenes involved primarily in developmental processes from GO categories for nucleic acid metabolism, DNA and RNA repair, and ribosomal biogenesis (Table 4). Thus, *S. graminum* appears to respond to the ingestion of phloem sap from 94M370 plants by repairing existing tissues while delaying immediate structural development. These delays in *S. graminum* structural development may explain why foliar damage was reduced on 94M370 plants but it is difficult to link components of the transcriptomes of *S. graminum* fed 94M370 plants to a lack of *S. graminum* population development.

*D. noxia*, a member of the Aphidini aphid tribe and a more specialized feeder on barley, oat, rye and wheat displayed completely different transcriptomes after ingesting phloem sap from 94M370 plants. When fed on either *Dn4* or 94M370 plants, *D. noxia* up-regulate unigenes involved primarily in detoxification and nutrient acquisition and down-regulate detoxification unigenes different than those upregulated, as well as unigenes involved in structural development. In contrast to the focus of *S. graminum* on DNA and RNA repair and delayed tissue growth, *D. noxia* appears to use a strategy of neutralizing the effects of the *Dn7* gene by induction of many more detoxification proteins and signaling proteins than *D. noxia* fed *Dn4* or *Dn0* plants.

Overall, the variation in transcriptional responses of *D. noxia* and *S. graminum* to the *Dn7* in 94M370 plants and to the *Dn4* resistance gene in Yumar plants suggests that the mechanisms underlying the evolution of virulent biotypes of these aphids are likely to be species-specific, even in cases where genes show some level of cross resistance.

## Supporting information

**S1 Table. Enriched gene ontology (GO) terms for all cluster 1 unigenes expressed at lower levels in *S. graminum* fed 94M370 plants compared to those fed either *Dn0* or *Dn4* plants.** DE = differentially expressed; FDR = false discovery rate.
(XLSX)

**S2 Table. Enriched gene ontology (GO) terms for all cluster 2 unigenes expressed at higher levels in *S. graminum* fed 94M370 plants compared to those fed either *Dn0* or *Dn4* plants.** DE = differentially expressed; FDR = false discovery rate.
(XLSX)

**S3 Table. Annotations of downregulated unigenes in *S. graminum* fed 94M370 plants.** SPROT match represents the annotation of the highest scoring blatsp match to the SWISS-PROT database.
(XLSX)

**S4 Table. Annotations of upregulated unigenes in *S. graminum* fed 94M370 plants.** SPROT match represents the annotation of the highest scoring blatsp match to the SWISS-PROT database.
(XLSX)

**S5 Table. Annotations of upregulated unigenes in *S. graminum* fed *Dn4* plants.** SPROT match represents the annotation of the highest scoring blatsp match to the SWISS-PROT database.
(XLSX)

**S6 Table. Annotations of downregulated unigenes in *S. graminum* fed *Dn4* plants.** SPROT match represents the annotation of the highest scoring blatsp match to the SWISS-PROT database.
(XLSX)

**S7 Table. Annotations of upregulated unigenes in *D. noxia* fed 94M370 plants.** SPROT match represents the annotation of the highest scoring blatsp match to the SWISS-PROT database.
(XLSX)

**S8 Table. Annotations of downregulated unigenes in *D. noxia* fed 94M370 plants.** SPROT match represents the annotation of the highest scoring blatsp match to the SWISS-PROT database.
(XLSX)

**S9 Table. Annotations of upregulated unigenes in *D. noxia* fed *Dn4* plants.** SPROT match represents the annotation of the highest scoring blatsp match to the SWISS-PROT database.
(XLSX)

**S10 Table. Annotations of downregulated unigenes in *D. noxia* fed *Dn4* plants.** SPROT match represents the annotation of the highest scoring blatsp match to the SWISS-PROT database.
(XLSX)

## Acknowledgments

The authors thank Drs. Subbaratnam Muthukrishnan, Yoonseong Park and Gregory Ragland for critical review of the manuscript.

## Author Contributions

**Conceptualization:** Lina Aguirre Rojas, Deepak Sinha, Charles Michael Smith.

**Data curation:** Lina Aguirre Rojas, Erin Scully, Laramy Enders, Deepak Sinha.

**Formal analysis:** Lina Aguirre Rojas, Erin Scully, Laramy Enders, Alicia Timm, Deepak Sinha.

**Investigation:** Lina Aguirre Rojas, Erin Scully, Alicia Timm, Deepak Sinha, Charles Michael Smith.

**Methodology:** Erin Scully, Laramy Enders, Deepak Sinha.

**Project administration:** Charles Michael Smith.

**Resources:** Charles Michael Smith.

**Software:** Erin Scully.

**Supervision:** Charles Michael Smith.

**Validation:** Erin Scully, Laramy Enders.

**Visualization:** Deepak Sinha.

**Writing – original draft:** Deepak Sinha, Charles Michael Smith.

**Writing – review & editing:** Lina Aguirre Rojas, Erin Scully, Alicia Timm, Deepak Sinha, Charles Michael Smith.

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
