## [Decision Letter · Decision Letter 0]

2 Jan 2020

PONE-D-19-24276

Comparative transcriptomics of Diuraphis noxia and Schizaphis graminum fed wheat
plants containing different aphid-resistance genes

PLOS ONE

Dear Dr. Smith,

Thank you for submitting your manuscript to PLOS ONE. After careful consideration, we
feel that it has merit but does not fully meet PLOS ONE’s publication criteria as it
currently stands. Therefore, we invite you to submit a revised version of the
manuscript that addresses the points raised during the review process.  In
particular:

Both
reviewers have indicated that more information is required about the
wheat lines that were tested, in particular how genetically similar (or
different) they are aside from the presence/absence of the resistance
allelesReviewer
2 also requested more information about genetic diversity in the aphid
populations tested, in particular as this relates to the phenotypic
variation observed on wheat lines.Both reviewers made cautionary statements related to
drawing conclusions based on annotations that are not functionally
validated.Consider
Reviewer 2's suggestion to provide a hypothesis in the Introduction as a
context for your Discussion and Conclusions, which may help
non-specialist readers follow the logic of the
paper.

You should also address all of the other recommendations of the
reviewers in your manuscript or rebuttal.

We would appreciate receiving your revised manuscript by Feb 16 2020 11:59PM. When
you are ready to submit your revision, log on to https://www.editorialmanager.com/pone/ and select the 'Submissions
Needing Revision' folder to locate your manuscript file.

If you would like to make changes to your financial disclosure, please include your
updated statement in your cover letter.

To enhance the reproducibility of your results, we recommend that if applicable you
deposit your laboratory protocols in protocols.io, where a protocol can be assigned
its own identifier (DOI) such that it can be cited independently in the future. For
instructions see: http://journals.plos.org/plosone/s/submission-guidelines#loc-laboratory-protocols

We look forward to receiving your revised manuscript.

Kind regards,

Owain Rhys Edwards, Ph.D.

Academic Editor

PLOS ONE

Journal Requirements:

1. Thank you for including the Introductory Note in your article to notify readers of
the previous article, the concerns regarding that work, and its retraction. Please
cite the retraction notice in that section, and include a full reference for the
retraction notice in the References section.

3. Please amend the manuscript submission data (via Edit Submission) to include
author Laramy Enders

4. Please include your tables as part of your main manuscript and remove the
individual files. Please note that supplementary tables (should remain/ be uploaded)
as separate "supporting information" files

Reviewers' comments:

Reviewer's Responses to Questions

**Comments to the Author**

1. Is the manuscript technically sound, and do the data support the conclusions?

Reviewer #1: Yes

Reviewer #2: No

2. Has the statistical analysis been performed
appropriately and rigorously? 

Reviewer #1: Yes

Reviewer #2: I Don't Know

3. Have the authors made all data underlying the
findings in their manuscript fully available?

Reviewer #1: No

Reviewer #2: Yes

4. Is the manuscript presented in an intelligible
fashion and written in standard English?

Reviewer #1: Yes

Reviewer #2: Yes

5. Review Comments to the Author

Reviewer #1: The authors investigated plant phenotypic responses and aphid gene
expression responses using three wheat cultivars(Yuma, Yumar, and 94M370) and two
aphid species (Schizaphis graminum and Diuraphis noxia). Most of the results section
consists of a listing of gene expression changes, along with speculation regarding
what the identified genes might be good for. There is no confirmation of candidate
gene function.

Specific comments:

1. “The Russian wheat aphid, Diuraphis noxia, (Kurdjumov) has invaded all wheat
producing regions in the world since being identified in 1900 and causes major
losses in yield of bread wheat, Triticum aestivum L.,(Quisenberry and Peairs 1998,
Baker et al. 2016). Based on recent CLIMEX projections, D. noxia distribution will
soon expand into Brazil, China, eastern North America, Northern Europe and New
Zealand (Avila et al. 2018).” These sentences seem contradictory. On the one hand,
D. noxia has invaded all wheat-growing areas and, on the other hand, there are
wheat-growing areas to which it will expand soon.

2. Some of the information under “Insect and Plant Material” is better placed in the
Results than in the Methods section. How the authors identified aphid species is the
method. The outcome of the identification process should be in the results
section.

3. The DNA sequencing results that confirm the aphid species should be deposited in
GenBank (or some other public database), and the corresponding GenBank ID number
should be included in the manuscript. This is particularly important given that the
authors have previously had issues with misidentification of aphid species.

4. There should be better description of the relatedness of the three wheat cultivars
(Yuma,

Yumar, and 94M370). How near-isogenic are they? It is mentioned that Dn7 is a
translocation from rye. How similar is the rest of the genome between the three
lines? In addition to the Dn4 and Dn7 regions, do other parts of the wheat genome
also influence plant symptom formation and aphid gene expression responses in these
experiments?

5. Lines 317-318: “which showed 95% and 100% nucleotide identity to Buchnera
aphidicola strains associated with S. graminum, respectively.” I think that this
sentence is missing “D. noxia”.

6. Lines 409-410. “All four GH family 1 unigenes have highest scoring blastp matches
to myrosinases, which are involved in degrading toxic glucosinolates produced by
plants in the order Brassicales.” Myrosinases do not degrade toxic glucosinolates.
Instead, they are involved in cleaving glucose off glucosinolates to produce toxic
breakdown products. It should also be noted that the myrosinase homology that the
authors found is not to plant enzymes, but to the myrosinase of Brevicoryne
brassicae (cabbage aphid), which sequesters glucosinolates and produces its own
myrosinase for activation as a defense against predators. Given that very few aphid
glucosidases have been functionally investigated, it is not surprising that the
closest identified match is to B. brassicae myrosinase. However, I would not read
any big meaning into this.

7. Figure 5 legend “clusters of highly expressed (2 - 8 fold) unigenes” This should
be “highly differentially expressed”, as the selection is based on fold-induction
rather than absolute expression level.

8. It would be helpful if the first line of each Excel supplemental table included
some sort of header stating what is shown in the table.

9. “Quality filtered reads from all nine S. graminum samples were pooled and a de
novo transcriptome

assembly was performed” and “All transcripts containing protein coding genes have
been submitted to NCBI’s Transcriptome Shotgun Assembly database (accession number
pending).” Why limit this to protein-coding assemblies? There is also information
content in the assemblies that were deemed to be non-protein-coding. I recommend
uploading all of the assemblies, as they might be useful in future analyses by the
authors and others.

Reviewer #2: The authors planned to conduct the comparative transcription analysis of
D. noxia biotype 1 and 2 feeding on three wheat cultivars, each contains a different
D. noxia resistance gene. However, they found later that “D. noxia biotype 2” was S.
graminum biotype I and reorganised the manuscript as submitted here.

Major problems

I had a hard time to read the manuscript because the manuscript does not contain any
hypothesis but contains a long description of the differentially expressed (DE)
genes and their GO categories. The authors tried to make a story out of the list of
DE genes, but to me, it looks like many things were happening in those aphids. In
case of D. noxia, it is not clear to me if they were simply dying or actively
responding to the plants. It is pity that the authors had to pool the samples from
different time points and missed the opportunity to dissect the DE genes. More
information of aphid performance (developmental time, survival rate, growth curve)
may help to understand the transcription data.

In the manuscript, there is not enough description of the wheat genotypes (it’s not
nice to let readers dig up references for such info), and no data was presented
regarding the interaction between D. noxia and the wheat genotypes: therefore, based
on this manuscript, readers cannot make their own hypothesis and examine the data
presented. I think that more information on the wheat genotypes (How were they
created? How genetically close are they?) and their interactions with D. noxia
biotype 1 (like Table 1 and also need to show survival rate and developmental time
etc) should be presented. The authors presented three wheat lines as “Dn0”, “Dn4”
and “Dn7”, but I think it is misleading the readers because these lines contain many
different genes in addition to the Dn genes. Especially, the Dn7 line was created by
transferring rye genomic segments (chromosome arms) into wheat cultivars. Both
tested aphids responded quite differently on Dn7 compared to Dn0 and Dn4, but the
differences might be caused by the rye genes in Dn7 wheat line and may not be
related to the Dn7 gene itself. I think this point should be clearly explained in
the manuscript. The subtitles like “Impacts of Dn7 on global gene expression in S.
graminum” is not correct. Especially, in case of Dn4, there was no impact of Dn4 on
wheat responses to S. graminum or on S. graminum fecundity, so such a subtitle is
simply confusing.

I also worry about their aphid stocks. Authors wrote “D. noxia biotype 1 aphids were
collected… (L110)” Those multiple aphids may have the same COI sequence, but they
may have many genetic differences… It is also not clear whether S. graminium biotype
I stock is a mixture of many S. graminum genotypes or not. The mixture of various
aphid lines (genotypes) may be a cause of the huge variation of aphid numbers
presented in Table 1.

Minor points

L42-44

Dn7 carrying wheat line had no effect on the fecundity of S. graminum, so, I don’t
agree with this sentence. What they saw in wheat (less chlorosis) should be
considered as a tolerance of the wheat and not a resistance reaction. Same for
L604.

L73-75: Although experimental support is lacking, “stealth” feeding behaviors is not
thought to mask aphid effectors from plant perception (at least, the cited
manuscripts don’t say that). Although effectors recognised by plant resistance genes
are not identified yet, infestation of specific aphids are recognised by these
resistance genes.

L79: In my knowledge, no one showed the variation in sensitivity of olfactory,
gustatory and salivary?! receptors. IAGC did not compare biotypes, and Smadja showed
high level of sequence differences in chemosensory genes in different A. pisum
biotypes. The sentence is incorrect.

Fig.4 legend, last sentence: cluster 2 instead cluster 1?

6. PLOS authors have the option to publish the peer
review history of their article (what does this mean?). If published, this will
include your full peer review and any attached files.

If you choose “no”, your identity will remain anonymous but your review may still be
made public.

**Do you want your identity to be public for this peer review?** For
information about this choice, including consent withdrawal, please see our
Privacy Policy.

Reviewer #1: No

Reviewer #2: No

---

## [Author Response · Author response to Decision Letter 0]

7 Feb 2020

February 6, 2020

Owain Rhys Edwards, Ph.D.

Academic Editor

PLOS ONE

Dear Dr. Edwards,

Below please find a complete set of author responses to reviewer comments of the
originally submitted manuscript PONE-D-19-24276. We believe that the revision
addresses reviewer concerns and meets reviewer suggestions. PONE style requirements
for file naming, formatting, table placement have been added, and all figure files
have been placed in the PACE digital diagnostic tool, confirmed and resubmitted to
PONE. 

In addition, the retraction notice includes a full reference for the retraction
notice, which is now in the References section. All transcripts containing S.
graminum protein coding genes have been submitted to NCBI’s Transcriptome Shotgun
Assembly database and all partial COI sequences used in D. noxia and S. graminum
identification have been deposited at Genbank.

Editor comments:

1. We note that you have stated that you will provide repository information for your
data at acceptance. Should your manuscript be accepted for publication, we will hold
it until you provide the relevant accession numbers or DOIs necessary to access your
data. If you wish to make changes to your Data Availability statement, please
describe these changes in your cover letter and we will update your Data
Availability statement.  

Author response: Raw sequencing reads have already been deposited under Bioproject in
NCBI’s Sequence Read Archive (SRA) under Bioproject PRJNA306025. SRA experiments
SRX1494436 to SRX1494443 and SRX1494451 are derived from S. graminum and SRX1494444
to SRX1494451, SRX1494434, and SRX1494435 are derived from D. noxia. Please see
lines 273-277 of the revised manuscript. Transcripts along with their corresponding
protein annotations have been deposited in NCBI’s Transcriptome Shotgun Assembly
database under XXXXX and the entire transcriptome assembly, including RNAs that were
not predicted to contain open reading frames are available at USDA’s Ag Data Commons
at the following url: https://doi.org/10.15482/USDA.ADC/1517669. In addition, partial
mitochondrial sequences for COI that were used to confirm the taxonomic identity of
each aphid species are deposited in GenBank under MT011383 for S. graminum I and
MN994435 for D. noxia biotype 1. 

2. Amend the manuscript submission data (Edit Submission) to include author
Enders

Author response: Done

3. Include the tables as part of your main manuscript and remove the individual
files. Supplementary tables (should remain/ be uploaded) as separate "supporting
information" files

Author response: Done

4. Upload figure files to the Preflight Analysis and Conversion Engine (PACE) digital
diagnostic tool, https://pacev2.apexcovantage.com/. to ensure that figures meet PLOS
requirements. To use PACE, you must first register as a user. Registration is free.
Then, login and navigate to the UPLOAD tab, where you will find detailed
instructions on how to use the tool. If you encounter any issues or have any
questions when using PACE, please email us at figures@plos.org. Supporting Information files do not need this
step.

Author response: Done

Reviewer 1 comments 

Comment: The Russian wheat aphid, Diuraphis noxia, (Kurdjumov) has invaded all wheat
producing regions in the world since being identified in 1900 and causes major
losses in yield of bread wheat, Triticum aestivum L., (Quisenberry and Peairs 1998,
Baker et al. 2016). Based on recent CLIMEX projections, D. noxia distribution will
soon expand into Brazil, China, eastern North America, Northern Europe and New
Zealand (Avila et al. 2018).” These sentences seem contradictory. On the one hand,
D. noxia has invaded all wheat-growing areas and, on the other hand, there are
wheat-growing areas to which it will expand soon.  

Author response: These sentences have been revised.

Comment: Some of the information under “Insect and Plant Material” is better placed
in the Results than in the Methods section. How the authors identified aphid species
is the method. The outcome of the identification process should be in the results
section.

Author response: As suggested, this information has been moved to the Results
section.

Reviewer comment: The DNA sequencing results that confirm the aphid species should be
deposited in GenBank (or some other public database), and the corresponding GenBank
ID number should be included in the manuscript. This is particularly important given
that the authors have previously had issues with misidentification of aphid
species.

Author response: Partial sequences for Schizaphis graminum I have been deposited at
GenBank under accession MT011383. Partial COI sequences for D. noxia biotype 1 have
been deposited at GenBank under accessions MN994435 (see lines 112-115). 

Reviewer comment: There should be better description of the relatedness of the three
wheat cultivars (Yuma, Yumar, and 94M370). How near-isogenic are they? It is
mentioned that Dn7 is a translocation from rye. How similar is the rest of the
genome between the three lines? In addition to the Dn4 and Dn7 regions, do other
parts of the wheat genome also influence plant symptom formation and aphid gene
expression responses in these experiments?

Author response: A detailed explanation of the pedigrees and relatedness of the three
wheat cultivars has been added to the revision (See Lines 126-135).   

Comment: Lines 317-318: “which showed 95% and 100% nucleotide identity to Buchnera
aphidicola strains associated with S. graminum, respectively.” I think that this
sentence is missing “D. noxia”. 

Author response: The authors thank the reviewer for determining this ommission. We
have revised this statement for additional clarity to read “which showed 95% and
100% nucleotide identity to 16s and 23s rRNAs derived from Buchnera aphidicola
strains associated with S. graminum, respectively.”(see line 366).

Comment: Lines 409-410. “All four GH family 1 unigenes have highest scoring blastp
matches to myrosinases, which are involved in degrading toxic glucosinolates
produced by plants in the order Brassicales.” Myrosinases do not degrade toxic
glucosinolates. Instead, they are involved in cleaving glucose off glucosinolates to
produce toxic breakdown products. It should also be noted that the myrosinase
homology that the authors found is not to plant enzymes, but to the myrosinase of
Brevicoryne brassicae (cabbage aphid), which sequesters glucosinolates and produces
its own myrosinase for activation as a defense against predators. Given that very
few aphid glucosidases have been functionally investigated, it is not surprising
that the closest identified match is to B. brassicae myrosinase. I would not read
any big meaning into this. 

Author response: The text regarding myrosinase has been revised to reflect the
reviewer’s concern. See Lines 548-553. 

Comment: Figure 5 legend “clusters of highly expressed (2 - 8 fold) unigenes” This
should be “highly differentially expressed”, as the selection is based on
fold-induction rather than absolute expression level.

Author response: As suggested, this text has been changed to “highly differentially
expressed” in the revision. 

Comment: It would be helpful if the first line of each Excel supplemental table
included some sort of header stating what is shown in the table. 

Author response: Each supplemental table now contains a title. 

Comment: Why limit this (S. graminum de novo transcriptome assembly) to
protein-coding assemblies? recommend uploading all of the assemblies…

Author response: Fully annotated protein coding unigenes have been submitted to TSA
under the accession XXXXXX (still awaiting # at revision submission) and a full
version of the assembly containing both protein coding and putative non-coding RNAs
is available at AgData Commons at https://doi.org/10.15482/USDA.ADC/1517669. 

Reviewer 2 comments 

Comment: I had a hard time to read the manuscript because the manuscript does not
contain any hypothesis but contains a long description of the differentially
expressed (DE) genes and their GO categories. The authors tried to make a story out
of the list of DE genes, but to me, it looks like many things were happening in
those aphids. In case of D. noxia, it is not clear to me if they were simply dying
or actively responding to the plants. It is pity that the authors had to pool the
samples from different time points and missed the opportunity to dissect the DE
genes. More information of aphid performance (developmental time, survival rate,
growth curve) may help to understand the transcription data.

In the manuscript, there is not enough description of the wheat genotypes (it’s not
nice to let readers dig up references for such info), and no data was presented
regarding the interaction between D. noxia and the wheat genotypes: therefore, based
on this manuscript, readers cannot make their own hypothesis and examine the data
presented. I think that more information on the wheat genotypes (How were they
created? How genetically close are they?) and their interactions with D. noxia
biotype 1 (like Table 1 and also need to show survival rate and developmental time
etc) should be presented. The authors presented three wheat lines as “Dn0”, “Dn4”
and “Dn7”, but I think it is misleading the readers because these lines contain many
different genes in addition to the Dn genes. Especially, the Dn7 line was created by
transferring rye genomic segments (chromosome arms) into wheat cultivars. Both
tested aphids responded quite differently on Dn7 compared to Dn0 and Dn4, but the
differences might be caused by the rye genes in Dn7 wheat line and may not be
related to the Dn7 gene itself. I think this point should be clearly explained in
the manuscript. The subtitles like “Impacts of Dn7 on global gene expression in S.
graminum” is not correct. Especially, in case of Dn4, there was no impact of Dn4 on
wheat responses to S. graminum or on S. graminum fecundity, so such a subtitle is
simply confusing.

Author responses: 

1. A paragraph describing previous research which documents responses of Dn0, Dn4 or
Dn7 plants to biotype 1 feeding and biotype 1 responses to these plants has been
added in lines 312-317.

2. A detailed explanation of the relatedness of the three wheat cultivars has been
added to the revision. See Lines 126-135.

3. Yes, the transcriptomes of both aphids fed Dn7 are different than those fed Dn0 or
Dn4. However, the phenotypic effects are similar. Dn7 plants sustain significantly
less foliar damage and lower populations of both species (significantly for D.
noxia, 36% for S. graminum) [see Tables 1 and 2]). It is highly unlikely that other
rye genes on the 1RS translocation function in aphid resistance, as Dn7 has been
mapped specifically to D. noxia resistance (see Anderson et al. Theor Appl Genet.
2003 107:1297, Lapitan et al. 2007). Further, several wheat pathogen defense genes
in the translocation linked to fungus, leaf rust, stem rust, stripe rust, and
powdery mildew resistance (Marais et al. 1994) have no role in D. noxia
resistance.

4. The authors respect the reviewer’s comments regarding the fact that there was no
impact of Dn4 on wheat responses to S. graminum. However, the subtitles were used to
show the impacts of the three plant resistance genes on both aphids. We believe the
reviewer’s confusion may be based on expectation of a gene(s) having a positive or
negative effect impact on an aphid. In this and any case of a comparison of effects,
a gene may have a positive effect, a negative effect or no effect.

5. Descriptive headers summarizing the results have been added to each subsection of
the results section to assist readers and help them make their own hypothesis of how
the data results were interpreted. 

Comment: Authors wrote “D. noxia biotype 1 aphids were collected… (L110)” Those
multiple aphids may have the same COI sequence, but they may have many genetic
differences.

Author response: The authors appreciate the reviewer’s concerns about potential
genetic differences in D. noxia biotype 1 but are perplexed about what differences
are being referred to. Nevertheless, D. noxia reproduces via parthogenesis which
minimizes any such differences. Second, all individuals used in experiments were
from a stock colony shown to be biotype 1 in annual plant differential diagnostic
test responses to Dn4 and Dn0 plants. 

Comment: It is also not clear whether S. graminium biotype I stock is a mixture of
many S. graminum genotypes or not. The mixture of various aphid lines (genotypes)
may be a cause of the huge variation of aphid numbers presented in Table 1. 

Author response: The authors also appreciate these reviewer concerns about genetic
differences in S. graminum biotype I but have equal concerns about the fact that the
reviewer does not specify what these differences are. Again, all individuals used in
experiments were S. graminum biotype I as determined by phenotype bioassay using
sorghum genotype Tx2783 which is sorghum breeding line TX2783, which is resistant to
biotype I and susceptible to biotype E (see line 306). 

Comment: L42-44 Dn7 carrying wheat line had no effect on the fecundity of S.
graminum, so, I don’t agree with this sentence. What they saw in wheat (less
chlorosis) should be considered as a tolerance of the wheat and not a resistance
reaction. Same for L604. 

Author response: Additional text has been added from lines 161- 168 to describe the
procedures used to measure tolerance in leaves of each genotype. These were the per
cent mean proportional dry weight change and the tolerance index, which removes the
potential bias of aphid population differences in tolerance measurements. In
addition, additional text has been added to lines 328-332 of the results to explain
the fact that although Dn7 plants had significantly less leaf damage and
proportional dry weight change than Dn0 plants, there were no differences in the
tolerance index between Dn0, Dn4, or Dn7 plants. Finally, the authors provide a
friendly reminder to reviewer of the comments in lines 771-780 stating that
tolerance, as measured by the plant tolerance index, could not be demonstrated, and
the suggestion of the possibility that that Dn7 plants may possess a resistance
mechanism unrelated to reduced plant tissue loss or reduced aphid population growth
that contributes to protection of foliar tissues from S. graminum-related
chlorosis.

Comment: L73-75: Although experimental support is lacking, “stealth” feeding
behaviors is not thought to mask aphid effectors from plant perception (at least,
the cited manuscripts don’t say that). Although effectors recognised by plant
resistance genes are not identified yet, infestation of specific aphids are
recognised by these resistance genes. 

Author response: The phrase “stealth feeding behaviors” has been replaced with
“release of suppressor proteins to mask aphid effectors.”

Comment: L79: In my knowledge, no one showed the variation in sensitivity of
olfactory, gustatory and salivary?! receptors. IAGC did not compare biotypes, and
Smadja showed high level of sequence differences in chemosensory genes in different
A. pisum biotypes. The sentence is incorrect.  

Author response: The IAGC and Smadja citations have been removed and replaced with
Eyres et al (2016), and the text accordingly revised.

Comment: Fig.4 legend, last sentence: cluster 2 instead cluster 1?

Author response: The figure 4D legend has been changed from cluster 1 to cluster
2.

On behalf of all of the coauthors, thank you very much for all of your assistance in
the processing and handling of this manuscript.

Kind regards, 

C. Michael Smith,

Fellow AAAS, ESA 

University Distinguished Professor

---

## [Decision Letter · Decision Letter 1]

20 Mar 2020

PONE-D-19-24276R1

Comparative transcriptomics of Diuraphis noxia and Schizaphis graminum fed wheat
plants containing different aphid-resistance genes

PLOS ONE

Dear Dr. Smith,

Thank you for submitting your manuscript to PLOS ONE. After careful consideration, we
feel that it has merit but does not fully meet PLOS ONE’s publication criteria as it
currently stands. Therefore, we invite you to submit a revised version of the
manuscript that addresses the points raised during the review process.

Both reviewers have responded similarly (and consistently to my own response) to the
additional information you have provided in this revision on the genetic relatedness
of the wheat lines tested.  You should address in your revision the two related
questions raised:

Is
Yuma an appropriate control plant for the 94M370 resistant line
containing Dn7?Can the DEGs observed between 94M370 and Yuma reasonably
be attributed to the Dn7 resistance gene? 

Neither reviewer believe this issue precludes publication of this research, but both
have indicated that the limitations of your design need to be made more explicit. 
You should also address the reviewers' other comments.

We would appreciate receiving your revised manuscript by May 04 2020 11:59PM. When
you are ready to submit your revision, log on to https://www.editorialmanager.com/pone/ and select the 'Submissions
Needing Revision' folder to locate your manuscript file.

If you would like to make changes to your financial disclosure, please include your
updated statement in your cover letter.

To enhance the reproducibility of your results, we recommend that if applicable you
deposit your laboratory protocols in protocols.io, where a protocol can be assigned
its own identifier (DOI) such that it can be cited independently in the future. For
instructions see: http://journals.plos.org/plosone/s/submission-guidelines#loc-laboratory-protocols

We look forward to receiving your revised manuscript.

Kind regards,

Owain Rhys Edwards, Ph.D.

Academic Editor

PLOS ONE

Additional Editor Comments (if provided):

The information provided by the authors on the genetic relationship among the wheat
genotypes tested provides some concerns about the interpretation of the results. The
first concern is that the best control plant has not been included for the
interpretation of the 94M370 line ("Dn7"). Previous studies have compared the
resistance phenotype (and its transcriptome) in 94M370 to Gamtoos as a near-isogenic
susceptible (Zaayman et al. 2009), the results of which are very relevant to the
submitted work and really should be referenced and discussed. The authors need to
defend why Gamtoos was not included in their experiments to provide a better control
for the 94M370 results.

The second (related) issue is that all the clustering results provided (visible in
Figs. 1, 3, 6, 7) shows quite clearly that there is more variability in
transcriptomic response associated with genetic background than with the resistance
phenotype. The authors have attributed this variation to the resistance provided by
Dn7, which is a legitimate hypothesis, but it is equally legitimate to attribute a
large part of this variation to the genetic background differences between 94M370
and the other two near-isogenic lines. It is not unreasonable for the authors to
propose their preferred hypothesis, but it needs to be presented with an appropriate
level of uncertainty and the alternative hypothesis should be given (especially
considering that there is a translocation containing additional genes, including R
genes, present in 94M370 that is absent from the other two lines).

The authors may need to reconsider referring to the lines as "Dn0", "Dn4", and "Dn7"
- as this may suggest to a reader who has not read the methods in detail that the
hosts are all near-isogenic, differing only in the presence/absence of these genes.
While this is the case for "Dn0" and "Dn4", it is definitely not the case for
"Dn7".

Reviewers' comments:

Reviewer's Responses to Questions

**Comments to the Author**

1. If the authors have adequately addressed your comments raised in a previous round
of review and you feel that this manuscript is now acceptable for publication, you
may indicate that here to bypass the “Comments to the Author” section, enter your
conflict of interest statement in the “Confidential to Editor” section, and submit
your "Accept" recommendation.

Reviewer #1: (No Response)

Reviewer #2: (No Response)

2. Is the manuscript technically sound, and do the data
support the conclusions?

Reviewer #1: Yes

Reviewer #2: Partly

3. Has the statistical analysis been performed
appropriately and rigorously? 

Reviewer #1: Yes

Reviewer #2: I Don't Know

4. Have the authors made all data underlying the
findings in their manuscript fully available?

Reviewer #1: Yes

Reviewer #2: Yes

5. Is the manuscript presented in an intelligible
fashion and written in standard English?

Reviewer #1: Yes

Reviewer #2: Yes

6. Review Comments to the Author

Reviewer #1: My comments and those of the other reviewer have been mostly
addressed.

Thank you for providing more information about the genetic background of the wheat
cultivars that were used. How genetically similar is Gamtoos, into which Dn7 was
crossed, to Yuma wheat. Is Yuma really the appropriate control for Gamtoos with a
1RS translocation from rye?

Lines 537-538. “All four GH family 1 unigenes have highest scoring blastp matches
to

538 myrosinases.” Please note at this point that these are myrosinases from cabbage
aphids, so that there is no confusion with the more commonly studied cruciferous
plant myrosinases. A reference describing cabbage aphid myrosinases would also be
appropriate here.

Reviewer 2 commented that this manuscript is difficult to read with a long listing of
differentially expressed genes, no hypothesis being tested, and no experimental
validation of the speculated function of differentially expressed genes. I agree.
However, this manuscript is being submitted to PLoS One, which does not require
results to be a significant advance over what was previously known.

Reviewer 2 questioned how genetically homogeneous the selected aphid populations are.
The authors’ response that they are all biotype 1 does not mean that they are
genetically homogeneous. Unless they started their colony with a single
parthenogenetic aphid, there is no easy way to determine that they all have the same
genotype. Or, if the different biotypes are known to be single parthenogenetic
clones, is there a reference for this?

Response to reviewers: “It is highly unlikely that other rye genes on the 1RS
translocation function in aphid resistance, as Dn7 has been mapped specifically to
D. noxia resistance” Based on information provided in the references, the Dn7 gene
has not been identified. Given the size of the current genetic mapping interval (2.4
cM and likely a few million bp of DNA), it is not yet possible to say that a single
gene affects aphid resistance.

Reviewer #2: I appreciate the information added by the authors, but I have the same
concerns as before. The three wheat genotypes (especially the one containing Dn7)
seem quite different from each other. The DE of the aphid genes feeding on those
three genotypes may reflect various physiological differences of the plants but may
not reflect the presence of Dn4 or Dn7. The manuscript concludes that “Overall, the
variation in transcriptional responses of D. noxia and S. graminum to the Dn7 and
Dn4 resistance genes suggests that the mechanisms underlying the evolution of
virulent biotypes of these aphids are likely to be species specific… (Line 777-)”,
but it should assume and discuss the limitation of their work.

7. PLOS authors have the option to publish the peer
review history of their article (what does this mean?). If published, this will
include your full peer review and any attached files.

If you choose “no”, your identity will remain anonymous but your review may still be
made public.

**Do you want your identity to be public for this peer review?** For
information about this choice, including consent withdrawal, please see our
Privacy Policy.

Reviewer #1: No

Reviewer #2: No

---

## [Author Response · Author response to Decision Letter 1]

24 Apr 2020

April 23, 2020

Owain Rhys Edwards, Ph.D.

Academic Editor

PLOS ONE

Dear Dr. Edwards,

Below please find author responses to reviewer and editorial comments regarding
PONE-D-19-24276R1 “Comparative transcriptomics of Diuraphis noxia and Schizaphis
graminum fed wheat plants containing different aphid-resistance genes.”

Both reviewers have two related questions to be addressed:

Q.1. Is Yuma an appropriate control plant for the 94M370 resistant line containing
Dn7?

Q.2. Can the DEGs observed between 94M370 and Yuma reasonably be attributed to the
Dn7 resistance gene?

Q.1. Author’s Response. We agree with the reviewers that the use of appropriate
controls is an important issue. The Dn4 and Dn7 genes have been previously linked to
D. noxia resistance in a previous study using Gamtoos wheat as a control (Zaymaan et
al. 2009 Physiol Plant 136: 209-222). The authors elected not to use Gamtoos as a
94M370 control because the Gamtoos 1RS arm contains the rust resistance genes Sr31
and Lr26, but the Turkey 77 wheat 1RS arm in 94M370 contains only Dn7 (reference 32
of current revision). The purpose of this current study was to determine whether two
lines carrying two different resistance genes for D. noxia also had resistance to S.
graminum. 

We have shown that Dn4 plants are susceptible to S. graminum and we have shown that
the wheat variety 94M370 containing the Dn7 gene displays some degree of resistance
against S. graminum. We suspect that this resistance could be conferred by Dn7, but
it is possible that the genetic background of Gamtoos or a combination of the
genetic backgrounds of 94M370 and Gamtoos could responsible for this resistance. We
have edited the manuscript to reflect this possibility (see response to Q2 below). 

Since our objective was to compare transcriptional responses of D. noxia and S.
graminum feeding on lines that had resistance to D. noxia only (Yumar) or both
aphids (94M370) and to determine whether 94M370 plants had similar effects on gene
expression in both aphid species, we believe that our differential expression
analysis and comparisons presented in the results section are valid. However, we now
refer to Dn7 as 94M370 to reflect the idea that differences in genetic backgrounds
between Dn0 and 94M370 could be responsible for some of the observed transcriptional
effects observed in both aphid species. 

Q2. Author’s Response. The reviewers and editor are correct that the DEGs observed
between 94M370 and Yuma could be influenced by differences in genetic background in
addition to the D. noxia Dn7 resistance gene. To address this concern, we have
revised the manuscript in lines 92-98 and lines 885-862 to reflect this possibility.
In addition, we also refer to plants containing the Dn7 gene as plants from wheat
genotype 94M370 instead of Dn7 plants as shown in the comment boxes throughout the
Revised Manuscript with Track Changes file and the titles of several figures and
Supplemental tables. We offer the dataset in the current revision as a first attempt
to understand Russian wheat aphid - greenbug comparative transcriptomics and refer
to them as "likely species-specific changes," allowing readers to develop a broad
interpretation of the results.

Additional Editor Comment: The authors may need to reconsider referring to the lines
as "Dn0", "Dn4", and "Dn7" - as this may suggest to a reader who has not read the
methods in detail that the hosts are all near-isogenic, differing only in the
presence/absence of these genes. While this is the case for "Dn0" and "Dn4", it is
definitely not the case for "Dn7".

Author’s Response. As indicated above, the revision includes reference to Dn7 plants
as 94M370 plants when discussing effects on S. graminum throughout the
manuscript.

As requested, the following items have been included in the revision of
PONE-D-19-24276_R1:

• A rebuttal letter responding to each point raised by the academic editor and
reviewers uploaded as a file labeled “Response to Reviewers.”

• A marked-up copy of the second revision to the manuscript highlighting the changes
in response to reviewer’s questions in PONE-D-19-24276_R1 uploaded as a file labeled
“Revised Manuscript with Track Changes.”

• An unmarked version of the second revision manuscript without tracked changes
uploaded as a file labeled “Manuscript.”

On behalf of all of the coauthors, we thank you very much for all of your assistance
in the processing and handling of this manuscript.

Kind regards, 

C. Michael Smith,

Fellow AAAS, ESA 

University Distinguished Professor Emeritus

to Reviewers.docx
---

## [Editor Report · Decision Letter 2]

29 Apr 2020

Comparative transcriptomics of Diuraphis noxia and Schizaphis graminum fed wheat
plants containing different aphid-resistance genes

PONE-D-19-24276R2

Dear Dr. Smith,

We are pleased to inform you that your manuscript has been judged scientifically
suitable for publication and will be formally accepted for publication once it
complies with all outstanding technical requirements.

With kind regards,

Owain Rhys Edwards, Ph.D.

Academic Editor

PLOS ONE
---

## [Editor Report · Acceptance letter]

11 May 2020

PONE-D-19-24276R2 

Comparative transcriptomics of *Diuraphis noxia* and
*Schizaphis graminum* fed wheat plants containing different
aphid-resistance genes 

Dear Dr. Smith:

I am pleased to inform you that your manuscript has been deemed suitable for
publication in PLOS ONE. Congratulations! Your manuscript is now with our production
department. 

With kind regards,

on behalf of

Dr. Owain Rhys Edwards 

Academic Editor

PLOS ONE